# A New Medical Analytical Framework for Automated Detection of MRI Brain Tumor Using Evolutionary Quantum Inspired Level Set Technique

**DOI:** 10.3390/bioengineering10070819

**Published:** 2023-07-09

**Authors:** Saad M. Darwish, Lina J. Abu Shaheen, Adel A. Elzoghabi

**Affiliations:** 1Department of Information Technology, Institute of Graduate Studies and Research, Alexandria University, 163 Horreya Avenue, El Shatby, Alexandria 21526, Egypt; adel.elzoghby@alexu.edu.eg; 2Department of Computer Information Systems, College of Technology and Applied Sciences, Al-Quds Open University, Deir AL Balah P920, Palestine; lina.shaheen@alexu.edu.eg

**Keywords:** brain tumor detection, medical images segmentation, bio-inspired clustering, level set segmentation, quantum-inspired computing

## Abstract

Segmenting brain tumors in 3D magnetic resonance imaging (3D-MRI) accurately is critical for easing the diagnostic and treatment processes. In the field of energy functional theory-based methods for image segmentation and analysis, level set methods have emerged as a potent computational approach that has greatly aided in the advancement of the geometric active contour model. An important factor in reducing segmentation error and the number of required iterations when using the level set technique is the choice of the initial contour points, both of which are important when dealing with the wide range of sizes, shapes, and structures that brain tumors may take. To define the velocity function, conventional methods simply use the image gradient, edge strength, and region intensity. This article suggests a clustering method influenced by the Quantum Inspired Dragonfly Algorithm (QDA), a metaheuristic optimizer inspired by the swarming behaviors of dragonflies, to accurately extract initial contour points. The proposed model employs a quantum-inspired computing paradigm to stabilize the trade-off between exploitation and exploration, thereby compensating for any shortcomings of the conventional DA-based clustering method, such as slow convergence or falling into a local optimum. To begin, the quantum rotation gate concept can be used to relocate a colony of agents to a location where they can better achieve the optimum value. The main technique is then given a robust local search capacity by adopting a mutation procedure to enhance the swarm’s mutation and realize its variety. After a preliminary phase in which the cranium is disembodied from the brain, tumor contours (edges) are determined with the help of QDA. An initial contour for the MRI series will be derived from these extracted edges. The final step is to use a level set segmentation technique to isolate the tumor area across all volume segments. When applied to 3D-MRI images from the BraTS’ 2019 dataset, the proposed technique outperformed state-of-the-art approaches to brain tumor segmentation, as shown by the obtained results.

## 1. Introduction

Brain tumors account for 85% to 90% of all primary central nervous system (CNS) tumors. Worldwide, an estimated 308,102 people were diagnosed with a primary brain or spinal cord tumor in 2020. In 2023, an estimated 24,810 adults (14,280 men and 10,530 women) in the United States will be diagnosed with primary cancerous tumors of the brain and spinal cord [1]. A brain tumor is the result of the development of malignant cells that have been allowed to continue unchecked either inside the brain or around the brain. Brain tumors are often divided into the following two broad categories: benign (noncancerous) and malignant (cancerous). Therefore, knowing the type of tumor that is present in the patient can be helpful in comprehending the patient’s situation. In the field of medicine, the early discovery and accurate classification of brain tumors are both extremely important skills to have. A correct prognosis made in a prompt manner is beneficial to the therapeutic process [2]. Brain biopsies and imaging techniques are often used to diagnose tumors. In open biopsies, a tiny hole is drilled in the skull, and a tissue sample is removed for microscopic examination of the tumor. This method poses serious risks. Medical imaging technologies improved detection by detecting tumors early and improving prognosis. An MRI is a three-dimensional scan of the brain, as shown in Figure 1. MRI scans now identify tumor type, size, and location. MRI can also distinguish soft tissue and spot tiny tissue density changes and tumor-associated metabolism variants [3,4]. Accurate MRI segmentation needs exact MRI image pixel marking. Segmentation aids brain tumor surgery and radiation therapy.

Image segmentation is a key area for development in the field of medical imaging. This study investigates automated contouring techniques as an alternative to the time-consuming process of manual contouring. The segmentation task makes an effort to pinpoint the location of the target by drawing a contour map of it. The complexity of the segmentation job increases as the number of visual differences between the objects of interest increases and as the number of irregular borders increases. Compared to 2D images, which only show one perspective, 3D images are more useful because they can show an object from every angle [5,6]. Direct automated segmentation of objects in 3D medical imaging is a challenging task since it often requires effectively identifying a large number of separate structures with complicated geometries inside a large volume under examination. Surface determination, where the boundary between one area and another is precisely captured, is a crucial notion in 3D image segmentation. The scan’s grayscale data are utilized to pinpoint precisely where these boundaries are. The exact procedure is highly variable, depending on the 3D image type and quality.

In response to these difficulties, the majority of modern machine learning strategies (e.g., deep learning) boost their learning potential by including more trainable parameters in their models. Since clinical imaging systems typically have low-end computer hardware with limited memory, CPU resources only, and a long inference time, increased model complexity will incur high computational costs and large memory requirements, making them unsuitable for real-time implementation on standard clinical workstations [1,2,6]. Consequently, due to the remote procedures, high-performance computing hardware such as high-end graphics processing units (GPUs) is typically required on remote servers to provide the large memory requirements for model training and to accelerate inference speed by segmenting MR examinations through parallel computation, which, because of the distant nature of the operations, are not easily accessible for use in real-time on clinical workstations. Moreover, trade-off techniques such as patch-wise or slice-wise training are often employed to fit a 3D dataset within these parameter-heavy models under limited computer memory, sacrificing fine-scale geometric information from input images and potentially affecting clinical diagnoses [1,2,3]. Since the MRI dataset has different spatial resolutions in the third dimension, 3D-MRI images are flattened into 2D slices.

Due to the overlap in intensity between tumor and normal tissue, the distortion of adjacent healthy tissues, and the prevalence of false positives, tumor identification is a challenging task because of the wide range of tumors in shape, location, size, and appearance features (see Figure 2). Many encouraging advances have been made in 3D brain tumor segmentation during the last decade, but there are still many pressing issues and obstacles to overcome. Large data sizes from high-resolution MRI pose a problem for 3D segmentation, necessitating substantial memory and processing resources [7,8]. Active contour modeling, a traditional segmentation method, can be used to segment MR images. In order to minimize the energy function as the active contour changed, they specified an energy function for it and sought out the ideal position [9]. In the literature, many algorithms employ a level-set approach for active contour image segmentation. While earlier methods struggled with geometry issues that arose during the evolution of curves, the level set approach proved to be an efficient solution [10].

The level set approach seeks to incorporate a curve into a surface. The level set approach offers a straightforward means of approximating the evolving structure’s geometry characteristics. Using a level set model has many benefits, including the ability to naturally and efficiently manage topological changes like joining and dividing and the depiction of contours in complicated topology. As a successful search strategy within the level set procedure, gradient descent is used to address this optimization issue. However, gradient descent techniques have a slow convergence rate and are sensitive to local optima [8,9,10,11]. While level sets have demonstrated a great potential for 3D medical image segmentation, their usefulness has been limited by two problems. First, 3D-level sets are relatively slow to compute. Second, their formulation usually entails several free parameters, which can be very difficult to correctly tune for specific applications. Figure 3 displays an example of level set segmentation of a brain tumor after adjusting the initial contour.

A number of recent articles have advocated for the widespread use of swarm algorithms in medical image segmentation [12,13,14]. One of the newest swarm optimization methods is the Dragonfly Algorithm (DA). It has been shown that DA converges to an optimal solution in many contexts [15]. Dragonflies, as part of their optimization process, should switch their focus from intensification to diversification to guarantee convergence. During the optimization process, the neighborhood area is increased to modify the flight route, and the swarm eventually comes together as a single unit to reach the global optimal. In comparison to other well-known evolutionary algorithms, such as the genetic algorithm and the swarm intelligence algorithms, DA has been proven to perform better [16,17,18,19,20]. As opposed to other swarm algorithms, DA can be relied upon and is easily included in preexisting optimization strategies. Convergence at a local optimum may, however, be caused by a lack of internal memory in DA. One of the issues a user might encounter with DA is the lack of connection between position updates and the algorithm’s clusters. Due to this trait, DA might get stuck in local optimums and be unable to find an optimal solution [19,20].

When applied to various swarm optimization techniques, the quantum computing mechanism (QCM) provides a revolutionary, intelligent method for quantamizing the searching behaviors of each individual. This is particularly true for the quantum rotation gate operation, which enables the user to choose a suitable rotation angle in their search for the optimal solution. Combining the QCM with soft computing methods has the potential to produce a new model in the fields of computer science and engineering. In this study, a quantum computing mechanism is used to quantamize dragonfly (QDA) behaviors in order to increase the searching effectiveness of the dragonfly algorithm by allowing each dragonfly to employ a quantum rotation gate to overcome the inertia weight during the search operations and to achieve the searching quality of Levy flying [21,22,23,24,25].

### 1.1. Research Motivation

Brain cancer is consistently ranked as one of the worst diseases in the world. So, the key to successfully treating cancer is an early diagnosis. The structural study of tumors in this area is challenging due to the complexity of the human brain. Current systems that make use of a variety of methods, such as threshold-based, model-based, and hybrid-based segmentation, have a number of drawbacks. These drawbacks include the fact that they take more time and are more prone to errors. The objective of this effort is to improve the physician’s perception of tumors in the brain. Level set methods are numerical approaches that provide extremely effective tools for analyzing and calculating interface motion in a variety of contexts. When used for medical image analysis and segmentation, the function labels each pixel or voxel, and optimality is set depending on desired imaging qualities. The level set approach is an excellent tool for modeling time-varying medical images and improving numerical calculations [8,9,10,11]. The disadvantage of level set methods is that they require a lot of planning to come up with appropriate velocities for advancing the level set function. Nonetheless, the payoff is a very versatile method for solving complicated issues over a broad spectrum [26]. In order to address the limitations of existing level set-based brain tumor segmentation systems, this paper reports on efforts to develop a model that combines quantum dragonfly algorithm-based clustering with the level set technique for segmenting 3D-MRI images of brain tumors.

### 1.2. Research Contribution and Novelty

In this work, a modified level set segmentation method based on a quantum-inspired optimization technique is suggested to detect tumors in 3D-MRI scans of the brain. The initial position is very important in managing topological variations in contours due to the tumor’s size, shape, and structural variability. To obtain the precise initial contour points for the level set segmentation method, the proposed approach employs an enhanced clustering strategy that integrates k-means and the quantum version of the dragonfly algorithm (QDA), in which the k-means are employed to determine the initial position of the QDAs population centroid, rather than using a random initial position. The preprocessing stage involves removing the brain from the cranium, and the two-step QDA is used to extract the tumor edges. As an initial contour for the MRI scan, these edges will be employed. Using a level set segmentation technique, the tumor location will then be detected from all volume slices. In this study, for optimization in the selection of the initial contour points, dragonfly behaviors are quantimized using quantum computing to improve the DAs search efficiency. To go from a locally optimal solution to a better solution (escape from the local solution) and hopefully to the global optimum, each dragonfly may employ a quantum rotation gate to overcome the inertia weight during the search operations and to conquer the searching quality of the Levy flight.

The remainder of the paper is organized as follows: Several recent works in this field are discussed in Section 2. Section 3 provides a detailed description of the proposed evolutionary level set segmentation technique using quantum dragonfly optimization. In Section 4, we provide an interpretation of our results using the 2019 BRATS dataset. In Section 5, the conclusions and future work are presented.

## 2. Related Work

Several different methods of brain image segmentation have been developed over the course of many decades in an effort to solve this optimization challenge. Researchers have also made significant progress in enhancing the effectiveness of segmentation algorithms. Complex medical image segmentation remains a challenging topic [1,2,3,27]. In the literature, current methods for segmenting brain images can be divided into the following four categories: intensity, atlas, deep learning, model, and hybrid-based segmentation. The advantages and disadvantages of these approaches are discussed in further depth in [28,29].

In [30], the authors propose a superior active contour model by fusing the level set strategy with the split Bregman technique. The energy function of this model, which includes the data fitting term and the length term, is supplied in a level set framework with neighbor region information for segmenting medical images. The minimization process is intended to be optimized by the variations in neighboring regions and local intensity. In order to minimize the energy function, the split Bregman method was applied, which resulted in faster convergence. After seeing promising results with their previous segmentation model, the authors decided to upgrade it by adding a multi-phase segmentation model and a 3D segmentation model specifically tailored to cardiac MR images.

To reliably and independently segment neighboring regions, the authors in [31] introduced a two-level set segmentation approach based on mutual exclusion. To avoid the creation of junction regions and ensure the independence of neighboring regions, the model makes use of two level set functions and uses the area of the joint region divided by the two level set functions to establish the mutual exclusion term of nearby areas. In order to construct previous constraints in the energy function, the approximate contours of the target area were manually established as prior knowledge. The authors obtained the level set formulation of this double-level set energy functional by including the data term, mutual exclusion term, prior constraint term, length term, and regularization term, and then minimized the energy using the gradient descent method.

In [10], the authors offered an automated method of sparse constrained level set segmentation for MR images of brain tumors. By analyzing images of brain tumors, this approach identifies the common features of the shape of brain tumors and creates a sparse representation model. An energy function based on the level set technique is produced by taking this model into account as a prior restriction. Their method outperforms the existing methods and achieves an average accuracy of 96.20 percent on MR images from the BraTS 2017 dataset.

The weighted level set model (WLSM) is presented in [32] for segmenting MR images with an inhomogeneous intensity that have been deteriorated by noise and weak boundaries. In order to precisely divide the interconnected regions of brain tissue, the authors first developed a weighted neighborhood information measurement strategy based on local multi-information and the kernel function. The level set’s sensitivity to initialization is reduced by using fuzzy c-means clustering’s membership function as the model’s spatial constraint; the level set’s function’s evolution may then be adaptively altered depending on input from different tissues. The distance regularization component of the level set function is replaced with a double potential function to keep the energy function stable throughout the evolution phase. However, this model does not analyze brain MRI images with diseases like multiple sclerosis or brain tumors. Furthermore, this study does not segment 3D MRI images directly but rather only explores the segmentation approach for 2D MRI slice images.

Restricted by clustering characteristics and the absence of a penalty term, segmentation accuracy quickly decreases as the number of tissues and the level of noise grow. In order to more effectively deal with these problems, the research presented in [33] suggested a level set technique that included a constraint term. First, the difference image is composed of T1- and T2-weighted magnetic resonance images, from which the cerebrospinal fluid atlas representing pre-segmented tissue and the white matter atlas utilized in the constraint term are derived. The T1-weighted image is, therefore, a representation of the real image multiplied by the bias field. A novel membership function under level set control is presented for the real image. Finally, energy minimization is used to simultaneously optimize components for bias field correction and contour evolution in tissue segmentation.

To combat individual anatomical diversity, intensity non-uniformity, and noise, which all have a negative impact on brain tumor segmentation, the work presented in [34] suggests an automated region-based strategy that combines fuzzy shape prior term with deep learning. By using an adaptively regularized kernel-based fuzzy C-Means clustering method, the authors of this study were able to create a unique energy function that incorporates tumor shape into the level set approach. This solves problems with standard level set techniques, including contour leakage and shrinkage. By utilizing U-Net to obtain the first contour, the method’s sensitivity to the initial contour selection can be reduced, making it totally automated.

An adaptive local variance-based level set (ALVLS) model was developed by the authors in [35] with the aim of segmenting MR images of the heart, MR images of the brain, and breast ultrasound images with intensity inhomogeneity and noise. The influence of the area term is adaptively modified by the ALVLS model based on the variance difference information. In order to increase the accuracy of medical image segmentation, the local intensity variances are optimized to maximize resistance to noise. For simultaneous segmentation of the left ventricles and left epicardium, they suggested using a two-layer level set model. The accuracy, efficiency, and noise resilience of the ALVLS model are all shown favorably in experimental results for both real-world medical images and synthetic images.

In [36], a brain tumor identification method for MRI images is recommended using an adaptive neuro-fuzzy inference system (ANFIS) classifier. The supplied RGB is first converted into a grayscale image. Using the skull stripping algorithm, non-brain tissues are removed during preprocessing. The resulting image is then segmented into tumor and edema using modified region growing and Otsu’s thresholding. Next, the wavelet coefficients, edge, and color histogram characteristics are recovered from the segmented “tumor portion” image. The grey wolf optimizer method is then used to choose the necessary characteristics from among the retrieved features. The adaptive-ANFIS then uses those characteristics to classify the tumor as either benign or malignant. The K-means clustering algorithm performs the adaptive process.

In [37], the authors introduced an active contour model based on adaptive weighted curvature, which combines heat kernel convolution with adaptively weighted high-order total variation to improve the efficiency of medical image segmentation for diagnosis. When approximating the boundary of a segmentation curve, the kernel convolution operation is used to reduce the computation complexity. To further emphasize local patterns and enhance segmentation precision, the weighted parameter in the high-order total variation term may be assessed automatically based on an adaptive input image. Two deep learning-based models for medical image segmentation were given by the authors in [38]. By combining transformer-based encoders with convolution-based encoders, their models made use of local and global features to accurately segment medical images. These models showed promise, outperforming the prior state-of-the-art in a number of different segmentation tasks, including brain tumors, lung nodules, skin lesions, and nucleus segmentation.

In order to include high-level organ shape information, the authors in [39] devised a contour-based annotation via an iterative deep learning technique that employs boundary representation rather than voxel labels. In order to enhance the precision of boundary identification, they implemented a contour segmentation network based on the extraction of features at several scales. They also created a human intervention technique based on contours to make it simple to change organ borders. Their system provides quick, few-shot learning and effective human checking by integrating the contour-based segmentation network with the contour-adjustment intervention technique.

Using a saliency map and deep learning feature optimization, the authors in [40] presented a computational strategy for brain tumor identification and classification. In the first stage, a contrast enhancement method based on fusion is presented. The next step is the presentation of a tumor segmentation method based on saliency maps, which is subsequently applied to the source images using active contour. The next step is to fine-tune and train a pre-trained CNN model in the following two ways: on improved images and on images of tumor localization. Features from the average pooling layer are used in the training of both models, which are trained through deep transfer learning. An enhanced fusion method called entropy serial fusion is then utilized to combine the deep learning features. The best features are chosen in the end using an improved dragonfly optimization method. The best features are then categorized using an extreme learning machine. The comparison to other neural networks demonstrates their framework’s advancement. Table 1 summarizes the discussed related work in terms of utilized techniques, advantages, and disadvantages.

Several different areas, including medical image analysis, password cracking, and pattern recognition, have taken an interest in quantum machine learning due to the impressive superposition and entanglement capabilities of quantum computing. While extensively utilized and showing promising results in medical image analysis, traditional machine learning still faces challenges, such as a lack of labeled data and inefficient processing. To address these issues, extensive research has integrated quantum computing and machine learning to investigate more sophisticated algorithms, which have shown notable gains in parameter optimization, execution efficiency, and error rate reduction. Understanding how quantum technology and medical image analysis interact is made possible by quantum machine learning, which also advances the field’s potential. Throughout the past decade, quantum machine learning has been increasingly put to use in the field of medical image analysis, and many studies have provided overviews of the field’s definition and categorization as well as an overview of the various quantum machine learning techniques and their applications [41,42].

### The Need to Extend the Related Work

According to the conducted survey, there are many problems with current 3D-MRI brain tumor detection methods [43,44,45,46,47,48,49,50]. These methods showed low accuracy when dealing with heterogeneous tumors and inaccurate segmentation results when dealing with noisy images. For thresholding-based segmentation approaches, the setting of the optimal threshold is very subjective. In cases of region-based segmentation, these methods require prior knowledge for parameter initialization and the post-processing step. The metaheuristics-based segmentation approaches are trapped in a local minimum due to an imbalance between exploration and exploitation. Furthermore, optimum representation feature determination is very subjective, just like parameter initialization. Deep learning-based segmentation approaches require high computational resources and high computation times, and in general, these models need complex network architecture. This review reveals that deep learning-based and hybrid-based metaheuristic techniques perform better than alternatives for accurate brain tumor segmentation. However, these approaches are inadequate because of their high computational and memory requirements.

Tumor detection in 3D MR images is complicated by issues including corner point generation, curve breaking, and combing; the level set approach, owing to its stability and irrelevancy with topology, offers a significant benefit in addressing these issues. This strategy does, however, have a number of drawbacks. Only objects whose edges are determined by the gradient may be segmented since the edge-stopping function is gradient-dependent. The curve may ultimately cross through object boundaries because, in reality, the edge-stopping function is never absolutely zero at the borders. It is mathematically required to maintain the evolving level set function near a signed distance function when using the level set approach. The development of a novel evolutionary quantum-inspired level set segmentation approach for a brain tumor detection system based on a 3D-MRI has, however, received little attention, as far as we are aware. The potential benefits of quantum machine learning include increased speed and accuracy, improved scalability, and a more efficient use of resources. Additionally, quantum algorithms can provide insights into data that would be difficult to uncover with classical algorithms [41,42].

## 3. The Proposed Evolutionary Quantum Brain Tumor Detection Method

Computer-assisted brain tumor diagnosis relies heavily on MR imaging-based tumor segmentation. Nevertheless, when using traditional segmentation techniques on MR images of the brain, unacceptable results are often produced due to factors such as inhomogeneous intensity, complicated physiological structure, and fuzzy tissue boundaries. To solve these problems, the proposed method employs the evolutionary quantum-inspired level set methodology to precisely define the tumor contour. The suggested approach does this by semantically fusing k-means with QDA-based clustering; that is, rather than employing a random center of the mass of the neighborhood (cluster centers), the k-means govern them appropriately. The cluster boundary points will be used to create an initial contour for the MRI series. The tumor region is then isolated across all volume segments using a level set segmentation approach. The primary model’s components and their interconnections are shown in Figure 4. The three primary phases of the model we propose for segmenting brain tumors are pre-processing, clustering using the QDA, and level set segmentation. The next sections elaborate on these phases.

### 3.1. Step 1 Preprocessing Phase

Due to the presence of noise in MRI scans, it is sometimes impossible to obtain reliable data for subsequent brain tumor detection and segmentation [51]. Enhancing image edges and, concurrently, removing or reducing noise incurred during acquisition and eliminating any inhomogeneous parts of the image that can lead to poor segmentation are two of the most important pre-processing activities for improving MRI quality. The suggested model used the following four procedures: 3D-MRI to 2D-slice conversion, skull-stripping, anisotropic diffusion, and contrast enhancement [52,53,54]. Open-source software platform 3D Slicer (https://www.slicer.org, accessed on 1 October 2022) is utilized for the conversion. It is used for medical image informatics, image processing, and 3D visualization. The following three morphological procedures are performed during the skull-stripping process: Otsu’s thresholding is used to transform the input 2D-MRI slices to binary images in the first step. Step 2 involves creating a mask from 2D MRI scans of the brain by diluting and eroding the slices. The brain becomes a connected, complete component by filling up the holes. As a third superimposed step, the final skull-strip image is generated by superimposing the mask over the original image (see Figure 5).

Noise may be reduced in images using a technique called anisotropic diffusion, which preserves important details like edges and lines without distorting the rest of the image (see Figure 6). Similar to the process that results in a scale space, anisotropic diffusion uses a diffusion algorithm to produce a parameterized family of gradually blurrier versions of an input image. The convolution of the input image with a progressively larger 2D isotropic Gaussian filter represents each of the output images in this series. The image is transformed into a linear and space-invariant form by the diffusion process. In anisotropic diffusion, the original image is combined with a filter that is itself dependent on the local content of the original image to obtain a family of parameterized images [53].

In the final step, contrast enhancement is applied by making use of the histogram equalization technique. This technique improves the visibility of tumors by rearranging the grayscale of the images in a non-linear fashion. Taking advantage of the physiological characteristics of human vision, this aids in the separation of subtle or obscured fluctuations in pixel intensity into a distribution that is more visually identifiable (see Figure 7). T1- and T2-weighted scans are the most typical MRI sequences. Time to echo (TE), the interval between the radio frequency (RF) pulse and the echo signal, and repetition time (TR), the interval between consecutive pulse sequences applied to the same slice, are both kept relatively short in order to generate T1-weighted images. The T1 characteristics of tissue are primarily responsible for the image’s contrast and brightness. T2-weighted images, on the other hand, are created with longer TE and TR periods. The T2 characteristics of the tissue are the primary determinants of contrast and brightness in these images. Figure 8 is a sample of 2D slice noise removal from a single patient.

### 3.2. Step 2 Quantum Dragonfly-Based Clustering Phase

Clustering is a data analysis method used to identify groups of data items within a dataset according to some measure of similarity [8,55,56,57]. The ultimate goal is to enhance detection performance by making it possible for the clustering model to accommodate a broad variety of brain tumor sizes, shapes, and structures (heterogeneous). Tumor boundaries are often unclear or irregular, with discontinuities. The identification of brain tumors has been a major focus of study in recent years, with a lot of attention paid to hybrid models [40]. The hybridization of artificial intelligence (AI) models with each other, with statistical models, and with meta-heuristic algorithms are just a few examples of the many AI applications that have received significant attention.

The third kind of hybridization, in particular, is very efficient [58]. Choosing the best algorithm to address a specific optimization problem is challenging since no one optimization method can be applied to all such issues. As a result, it is crucial to combine them with meta-heuristic algorithms in order to compensate for their shortcomings and choose appropriate combinations of clustering model parameters. The dragonfly algorithm (DA) is a newly suggested metaheuristic optimization algorithm that takes its cues from the complex behaviors of dragonfly swarms [20,21]. DAs main benefits lie in its relatively few control parameters and its simple, adaptable, and easily applied structure [40]. Many optimization problems in medical image analysis have been solved by employing DA [59,60,61,62,63,64]. Convergence is slow, and the algorithm gets stuck in local optima because the DAs constrained design hinders monitoring the best searching experience of dragonflies in earlier generations throughout the local searching phase [65]. This study quantamize dragonfly behaviors using a quantum computing technique, creating a new algorithm called Quantum Dragonfly (QDA) that improves the dragonfly algorithm’s search efficiency, as illustrated in Figure 9 [21,22]. Since the QDA has never been hybridized with the clustering model, this study focuses on combining the DA and the QCM with a clustering model in order to address its limitations.

In our model, the AD technique for clustering issues is enhanced by applying the k-means algorithm and quantum computing to speed the convergence rate while preserving the balance between exploration and exploitation. Here, K-means is utilized to construct the correct centers to deal with the randomization to determine the center of mass of the neighborhood inside the traditional DA algorithm. In addition, the concept of a quantum rotation gate is used to increase the DA swarm’s variety and optimize the swarms’ fitness by sending them to more productive areas. The combined effects of these factors provide a more stable balance between the QDA-based clustering method’s diversification and intensification capabilities.

#### 3.2.1. Initial DA’ Food Sources Extraction Using K-Mean Algorithm

In this step, the DA’ initial food sources (the neighborhoods’ mass centers) are located using the K-means method. In our case, four clusters are recognized here that stand in for the cerebrospinal fluid or background, grey matter, white matter, and tumor (see Figure 10 and Figure 11) [66,67]. Then, the closest centroid is selected for each point. Assigning data points to the centers of the closest clusters requires some kind of proximity measure. In this study, we adopt the standard Euclidean distance as our proximity metric. After the clusters have been formed and all the points assigned to them, the centroids are recalculated using the cluster’s mean. Assigning points to clusters is performed again and again until there is no movement across clusters [57]. Figure 12 shows the main steps of the k-mean clustering procedure.

#### 3.2.2. Determine Initial Contour Points Using Quantum Dragonfly Algorithm

In this step, an effective clustering technique is implemented after the k-means algorithm is used to regulate the clusters’ centers (cluster head selection). The clustering that is utilized to group and establish the initial tumor contour points is optimized using, in our case, the quantum-inspired dragonfly algorithm to boost accuracy. The DA method is based on both dynamic (migration) and static (hunting) swarming particles. When it comes to optimization, these two swarming techniques rely on the usage of metaheuristics for both exploitation and exploration [68]. Figure 13 illustrates the flowchart of the Dragonfly algorithm [21,69,70]. Separation ($S_{i}$), alignment ($A_{i}$), cohesion ($C_{i}$) attraction to food sources ($F_{i}$), and distraction from enemies ($E_{i}$), are the five operators of dragonfly swarms [21]. First, the $S_{i}$ operator distinguishes the *i*th dragonfly’s static condition from that of its neighbors. The $A_{i}$ operator makes sure that the speed of the *i*th dragonfly is the same as the speed of all the other dragonflies in the area. When the *i*th dragonfly does a search, $C_{i}$ operator directs its attention to the spatial center of the immediate area (neighborhood). The *i*th dragonfly’s primary survival goal is to collect food, but it is diverted from this goal by enemies, as reflected by the final two operators, $F_{i}$ and $E_{i}$. To show the survival behaviors of dragonfly swarms, each operator is weighted appropriately [70].

The Lévy flight, a kind of random walk in which the step lengths form a Lévy distribution that is heavy-tailed, is used to update a dragonfly’s location if it is unable to find a neighbor within its searching radius after multiple rounds. If you think of it as walking across a place with more than one dimension, you will notice that your steps go in completely random, isotropic directions. Position and velocity updates are used to determine fitness values (segmentation error). Up until the stopping criterion is met, the position is updated. The smallest fitness value (segmentation error) and its corresponding location represent the optimal solutions [71]. Both the inertia weight and the Lévy flight equation have their origins in Newtonian mechanical concepts, which place limits on the kinds of searching that may be performed by an individual. Any given dragonfly has a high degree of difficulty altering its preexisting searching direction in response to DA because of the so-called inertia constraint; any dragonfly without an adjacent neighbor must use the Lévy flight equation and may not modify the fly rules while in Lévy flight due to the Lévy flight constraint [69,70,71].

In order to overcome the dilemma of the dragonfly individual, the QCM (especially for different quantum rotation angles, $\theta_{i}$, as illustrated in Figure 9) might be used to substitute these two searching behaviors (inertia behavior and Lévy flying behavior). Therefore, a dragonfly may escape from a local optimal solution and, with any chance, arrive at the global optimum. Local optimal solutions (Local optimal 1, 2, and 3) may be reached by a dragonfly at any given time throughout its searching trip in any of numerous directions, depending on the flyer’s inertia search directions or Lévy flying constraints. To overcome the dragonfly’s dilemma, QCM with a range of quantum rotation angles ($\theta_{i}$) is used to find the best solutions (1, 2, 3, etc.) and, ultimately, the global optimal solution. The details of the utilized QDA are as follows [21,22,23,24,25,72,73]:

Step 1 Initialization: randomize the dragonfly population$\;X_{i}=\left(x_{i}^{1},x_{i}^{2},\ldots .,\;x_{i}^{d}\right),\;i=1,2,\ldots .,N$ of size N and *d* represents the dimension of the *i*th dragonfly (*d* = 1 in our case) for the variable of our clustering model (one parameter represents the image pixel$\;X_{i}^{p}$). Each dragonfly individual’s fitness value is calculated with their initial position, which is often created at random between the minimum and maximum values of the MRI image pixel. In this study, we define fitness as the standard deviation of the clustering errors produced by a model trained using the identified tumor pixel. The following equations are used to determine a dragonfly individual’s weights for the five operators listed above based on their initial values, which are determined at random.
(1)$$S_{i}=-\sum_{J=1}^{M}X_{i}-X_{j}$$
(2)$$\;A_{i}=\frac{1}{M}\sum_{J=1}^{M}V_{j}$$
(3)$$C_{i}=\frac{1}{M}\sum_{J=1}^{M}X_{i}-X_{j}$$
(4)$$F_{i}=X^{food}-X_{i}$$
(5)$$E_{i}=X^{enemy}+X_{i}$$
(6)$$\;D_{ij}=\sqrt{\sum_{k=1}^{d}{\left(x_{i}^{k}-x_{j}^{k}\right)}^{2}}$$

In these equations, $X_{j}$ and $V_{j}$ denote the position and velocity of the *j*-th neighboring individual; $X_{i}$ denotes the position of the current individual; *M* denotes the number of neighboring individuals; $X^{food}$ and $X^{enemy}$ denote the locations of the food and enemy sources, respectively;$\;D_{ij}$ is the Euclidean distances between all pairs of dragonflies used to verify the individual neighbors with size *M*.

Step 2 Fitness evaluation: From the quantimized positions of the proposed model’s parameter, evaluate the fitness values (in this work, clustering accuracy indices are utilized as fitness values). Clustering accuracy is measured using a metric called the mean absolute percentage error (MAPE).
(7)$$\mathrm{MAPE}=\frac{1}{N}\sum_{i=1}^{N}\left|\frac{y_{i}-f_{i}}{y_{i}}\right|\times 100\%$$

N is the total number of clustering results; $y_{i}$ is the actual class of the instance *i*; $f_{i}$ is the predicated class of the same instance.

Step 3: Dragonfly operators updated: For each parameter population, calculate the five operators, $S_{i}$,$\;A_{i}$,$\;C_{i}$, $F_{i}$, and $E_{i}$. Then, update the value of each operator according to.
(8)$$\;X_{t+1}=X_{t}+\Delta X_{t+1}$$
(9)$$\Delta X_{t+1}=\left(sS_{i}+aA_{i}+cC_{i}+fF_{i}+eE_{i}\right)+\xi \Delta X_{t}$$

*s*, *a*, and *c* represent the weights of separation, alignment, and cohesion, respectively; *f* and *e* represent the factors of food and enemy, respectively; *ξ* represents the inertia weight, and *t* is the number of the current iteration.

Step 4: Inertia motion replaced by QCM: Quantamize $\Delta X_{t}$ into the qubit format$\;q_{k}$ using the following equation in order to update$\;\xi \Delta X_{t}$:(10)$$\;q_{k}=\frac{2\Delta X_{t}-\left(max+min\right)}{max-min}\;,\;k=X_{i}^{p}$$

*max* and min represent the upper and lower bound on parameter$\;X_{i}^{p}$, respectively. In this case,$\;q_{k}$ can be reorganized in qubit format, *Q*. The smallest unit in a quantum system is called a quantum bit, or qubit, and it may be in either the “0” or “1” state, or in any superposition of the two [41,42]. Therefore, the state of a qubit is given by the following:(11)$$\left|\psi \rangle =\beta_{1}\right|0\rangle +\beta_{2}|1\rangle$$
(12)$${\left[\beta_{1}\right]}^{2}+{\left[\beta_{2}\right]}^{2}=1$$
where |0〉 and |1〉 are the values of traditional bits 0 and 1, respectively; $\beta_{1}$ and $\beta_{2}$ are the probabilities associated with their corresponding states. The linear superposition of all possible states in a system with *k* qubits and 2*^k^* states may be written as a generalization of this formula.
(13)$$|\psi_{i}\rangle =\sum_{i=1}^{2^{k}}p_{i}|S_{i}\rangle$$

In which the probability associated with state$\;S_{i}$, denoted by$\;p_{i}$, is normalized to 1.
(14)$${\left[p_{1}\right]}^{2}+{\left[p_{2}\right]}^{2}+\ldots +{\left[p_{2^{k}}\right]}^{2}=1$$

One qubit out of *k* qubits has probability
(15)$$q=\left[\left.\begin{array}{c} \alpha_{1}\\ \beta_{1} \end{array}\right|\left.\begin{array}{c} \alpha_{2}\\ \beta_{2} \end{array}\right|\ldots .\left|\begin{array}{c} \alpha_{K}\\ \beta_{k} \end{array}\right.\right]\;,{\left[\alpha_{i}\right]}^{2}+{\left[\beta_{i}\right]}^{2}=1\;,\;i=1,2,\ldots k$$

One helpful operator for modifying qubits is the quantum rotation gate. Using matrices, the procedure may be implemented. The quantum rotation gate with phase angle θ is used to improve the current state’s solution.
(16)$$\overset{´}{P}=\left[\begin{array}{cc} \cos\theta & -\sin\theta \\ \sin\theta & \cos\theta \end{array}\right]P$$

$\overset{´}{P}$ is the updated position, and *θ* is the given angle of the quantum rotation gate (see Figure 14)

Step 5 New solution generated by quantum rotation gate: To obtain a different value of *Q*, *Q*’, depending on a particular phase angle, use the quantum rotation gate (Equation (16)). Then, carry out the dequantization process to produce a real number format for each corresponding new q in accordance with the following:(17)$$\Delta X_{t}^{\prime }=\frac{1}{2}\left[max*\left(1+q^{\prime }\right)+min*\left(1-q^{\prime }\right)\right]$$

Step 6 Position updated: Plug the new value of $\Delta X_{t}^{\prime }$ into Equation (9) to obtain the value of $\Delta X_{t+1}^{\prime }$. Finally, to reflect the current distribution of dragonflies, we may plug $\Delta X_{t+1}^{\prime }$ into Equation (8). It is therefore necessary to return to Step 2 and calculate the fitness values (clustering errors) based on the quantified value of $\;X_{i}^{p}$.

Step 7 Lévy flight replaced by QCM: To update the Lévy flight behavior, use the QCM to compute
(18)$$q=\frac{2X_{t}-\left(max+min\right)}{max-min}$$

Furthermore, use quantum rotation gate operation to obtain $X_{t}^{\prime }$ in order to update the positions of dragonfly populations.
(19)$$X_{t+1}^{\prime }=X_{t}^{\prime }+levy\left(d\right)X_{t}$$

*t* is the number of the current iteration, and *d* represents the number of dimensions in which the dragonfly’s position is specified (*d* = 1). This improves the probability of jumping out of local optimum and finding a globally optimum solution, i.e., to expand its search space. Go back to Step 2 to evaluate the fitness values from the quantamized positions of $\;X_{i}^{p}$.

Step 8: Gaussian mutation-based DA: DA is not efficient enough in both the exploration and the exploitation phases. Therefore, it needs to be more diffused to search for more spaces to achieve the diversity of the population. In our case, the Gaussian mutation mechanism is employed to increase its diversity.
(20)$$X_{t}^{\prime }=X_{i}+\left(1+k\times randn\left(1\right)\right)$$

$X_{i}$ is the *i*th individual of the current population, $X_{t}^{\prime }$ is the new individual obtained after Gaussian mutation, k is the weight parameter (*k* = 1, as stated in [22]), and randn (1) produces 1 × 1 matrix; that is, a data from 0 to 1.

Step 9 Stop criteria: The optimal solution is the optimal set of clustering model parameters if the number of iterations approaches the threshold; otherwise, the procedure loops back to Step 2 and continues searching.

### 3.3. Step 3 Level Set Segmentation

Partial differential equations (PDE), which involve progressively evaluating the differences between neighboring pixels to determine object boundaries, provide the basis of a large class of contemporary image segmentation algorithms, among which level sets are an important subclass [74]. The method should ideally converge to the object’s edge, as that is where the differences are greatest. Level sets advance a contour like a rubber band until the contour hits an object boundary. If there are any gaps in the border, the contour is prevented by its rubber band-like shape (=curvature). A grayscale difference and the rubber band’s strength may be pre-selected [75]. Up until it reaches a predetermined difference in the intensities of the pixels, the rapid marching algorithm expands from a seed point extracted from the previous step to the object boundary, as illustrated in Figure 15.

Using a variety of inherent geometric measurements of the image, active contours develop an initial contour over time. The steps used by the plugin implementation include an edge-based constraint, a grey-value penalty, and a curvature constraint, all of which serve to avoid leakage of the object boundary in poorly edged regions. Due to the implementation’s active contours’ ability to split and merge during curve evolution, it may be utilized to identify several objects. The method used in the plugin is state-of-the-art; it is memory-efficient, runs quickly, and can be simply adapted to accommodate different level set algorithms [76]. This implementation is based on the following PDE updates [77,78]:(21)$$\Phi \left(i\right)=\Delta Tg\left(i\right)\left(W_{a}F_{a}\left|\nabla \Phi \right|+W_{c}F_{c}\left|\nabla \Phi \right|\right)$$
(22)$$g\left(I\right)=\frac{1}{1+\left(\nabla I^{*}+g\right)\cdot 2}$$
(23)$$\Delta T=\frac{1}{6\cdot W_{a}\cdot W_{c}}$$

$\Phi \left(i\right)$ is the ISO surface at the current iteration *i*, $\nabla I^{*}$ is the difference in the smoothened image. An ISO surface is a three-dimensional analog of an ISO line. It is a surface that represents points of a constant value within a volume of space; in other words, it is a level set of a continuous function whose domain is 3-space. $W_{a}$ and $W_{c}$ represent advection and curvature weight, respectively, while $F_{a}$ and $F_{c}$ represent advection and curvature force, respectively. A more detailed explanation of the algorithm can be found in [79,80,81]. Figure 16 illustrates the level set steps in medical image segmentation [82]. According to the moment, the contour’s acceleration and velocity change throughout segmentation. The velocity and acceleration rise as the contour $\Phi \left(x,t\right)$ gets farther away from the solution. The acceleration and velocity drop as the contour gets closer to the target region. Figure 17 shows the result of applying the level set segmentation to a set of images.

In conclusion, the level set method makes it very easy to follow shapes that change topology, i.e., it is a great tool for modeling time-varying objects. However, the standard level set algorithm requires a significant amount of computation and a large number of iterations if the initial contour deviates much from the actual contour. In our case, two-step quantum behavior DA clustering brings the initial contour set closer to the actual tumor contour, allowing for excellent convergence in fewer iterations. The main argument in favor of this approach is that it may construct a more trustworthy tradeoff between the exploration of all search regions and the exploitation of the proximity of solutions. It begins each cycle with a rapid increase in the rate at which it generates exploratory trends and then gradually shifts to a smooth transition between exploration and exploitation. As a consequence of the smooth transition and consistent performance, the quality of the outcomes has increased, and the risk of stagnation has decreased. Mutation operations may assist individuals who are escaping poor-quality solutions in a situation of immature convergence. In QDA, a wave function expresses the particle’s motion state, and the Schrodinger equation in quantum mechanics yields the corresponding probability. All the particles have a chance of showing up in the search space, and the population’s capacity for worldwide exploitation is greatly enhanced as a result.

## 4. Experimental Results

The performance of the proposed model is validated in this section using MRI volumes from 285 patients who have brain tumors, each of which represents a distinct tumor type, location, size, and intensity. The 3D-MRI images of brain tumors are available in the BraTS 2019 (Brain Tumor Segmentation) test and challenge dataset [83,84]. The experiment was run in MATLAB R2018a on a computer with an Intel (R), Core (TM) i3 CPU, and 8.00 GB of RAM. Specifically, this context makes use of accuracy, recall (sensitivity), precision, dice score, Hausdorff 95, and specificity as evaluation metrics. To verify the effectiveness of the proposed model, we conducted the first set of experiments using the BraTS 2019 dataset and compared it to the state-of-the-art brain tumor segmentation techniques given in Table 2. A brain symmetry analysis technique and a combination of region-based and boundary-based segmentation approaches form the basis of the model presented in [85]. The limitation of region segmentation is that it can only be used with closed boundaries. In [86], the authors use a combination of region-based K-means clustering and variational level sets to fix the issue of poor convergence towards the concavities of the tumor border. The K-value is notoriously unpredictable, and various starting partitions might provide wildly varied clustering results.

Moreover, models from [59,87,88] were selected to compare the proposed model to other well-known methods for segmentation. In [87], MRI scans are prepared for tumor segmentation using a level set technique with certain modifications. It is also crucial to extract informative characteristics in order to make reliable predictions about the image’s class. An artificial neural network that can learn from its errors is used to make the classifications. The adaptive ANN uses the whale optimization method to fine-tune the performance of the layer neurons. However, neural networks are complex and require a lot of data to train them. The method suggested in [88] uses a level set evolution based on the minimization of an objective energy functional, the components of which are weighted according to their relative importance in boundary detection. Local edge characteristics gathered from neighboring areas within and outside the evolving contour are used to calculate this relative relevance. Forcing the level set function to be near distance function using a variational formulation, however, requires more extracted information and, therefore, a higher computational cost. The proposed model is an expansion of the model given in [59], which uses a quantum-inspired dragonfly algorithm rather than a conventional dragonfly method to reliably extract initial con-tour points prior to using a level set segmentation technique.

The proposed model is validated by the results, which show a 2.5% improvement in accuracy compared to the Dragonfly, Level set-based brain segmentation technique. The value of the Dice score can exceed 0.95 for whole tumor segmentation, showing good overlap with manual segmentations. In terms of specificity, the ratio is equal to 0.993 for the tumor region. So, it means that the segmentation results are reliable enough. One potential reason for this result is that a two-step QAD identifies a more accurate initial contour than the comparison approaches do, thereby increasing the segmentation accuracy. Information gathered from k-means is used to support the QDAs search engine. By using a quantum-inspired computing paradigm, the suggested model is able to stabilize the exploitation/exploration trade-off and make up for any weaknesses of the conventional DA-based clustering approach, such as delayed convergence or being stuck in a local optimum. Furthermore, the suggested model has a robust local search capacity by adopting a mutation procedure to enhance the swarm’s mutating and realizing its variety. Figure 18 shows a sample of segmentation results.

Another set of experiments was carried out to compare the proposed model with recent works [89,90,91], as shown in Table 3, to validate the proposed model against existing approaches that use deep neural networks (DNNs), one of the most well-known techniques for segmenting brain tumors. The first approach uses variational level sets (VLS) in combination with deep learning to deal with the level set’s sensitivity to initial settings; it depends on the number of iterations. In the second approach, a unique symmetric deep convolutional neural network is suggested for autonomous brain tumor segmentation. By adding symmetrical masks at many layers, it enriches segmentation networks based on deep convolutional neural networks (DCNN). The third strategy is based on a fusion support vector machine technique for deep convolutional neural networks. The results in terms of accuracy, recall, and precision reveal the segmentation power of the suggested model compared with other DNN-based methods. This indication has been verified through other evaluation metrics such as dice score (The dice metric measures volumetric overlap between segmentation results and annotations) and specificity (evaluates a model’s ability to predict true negatives of each available category), even if the results are more or less in agreement with these approaches. Overfitting is the primary issue with CNN, and the need for a large training set makes it computationally costly.

Despite the fact that quantum clustering and the level set technique are not particularly novel to the area of brain tumor segmentation, an additional series of experiments were performed to validate their effectiveness when combined. The suggested model’s QDA component has been swapped out with a set of well-known quantum-inspired metaheuristic modules running in black box mode with their initial settings preserved. Quantum Particle swarm optimization (QPSO), Quantum artificial bee colony (QABC) algorithm, and Quantum clown fish (QCF) method are all examples of employed quantum-inspired metaheuristic algorithms [92,93,94]. The results shown in Table 4 support the study’s hypothesized improvement in segmentation accuracy when utilizing the QDA classifier based on correct center points (initial contour seeds) retrieved using k-means. When compared to the closest combination between the QCF and the level set segmentation, the proposed combination improved accuracy by at least 1%. Whether a QPSO, QABC, or QCF algorithm is more exploratory or exploitative depends on the convergence rate, which is, in turn, determined by the major factors that affect the individual’s movement toward the best position found so far. The convergence rate in QDA is parameter-free. The suggested model has a robust local search capacity by adopting a mutation procedure to enhance the swarm’s mutating and realizing its variety.

The next set of experiments is being conducted to prove that k-means has a significant impact on improving segmentation precision. Table 4 demonstrates the efficacy of the proposed model whereby k-means is used to calculate the initial DA population; compared to the model with random initialization of QDA, it improves accuracy by 11%. Instead of using random initialization, the k-means algorithm is employed in this study to establish the appropriate locations in a random search space for food sources; therefore, the results of the DA clustering algorithm are enhanced with this method. In this scenario, the initial food source placements in the DA algorithm are determined by the centers of the clusters produced by k-means. Table 5 shows that the computed standard deviation is small, which means that the data points cluster tightly around the mean, and there are no extreme outliers. This indicates that the model’s accuracy metrics are robust over a wide range of MRI images. Figure 19 shows a sample of the segmentation process using the quantum DA-based level set segmentation procedure. It can be deduced that the QDA method significantly decreases the required number of level set iterations. The contour was calculated very closely to the tumor region using the best-configured two-step QDA parameters.

To confirm the efficiency of the suggested model to segment different types of tumors with multiple tumor lumps per slice in different planes, the last set of experiments was conducted using a benchmark brain tumor dataset from the website https://figshare.com/articles/dataset/brain_tumor_dataset/1512427/5 (accessed on 1 October 2022). In this dataset, there are the following three types of tumors: meningioma (708 images), glioma (1426 images), and pituitary tumor (930 images). All images were acquired from 233 patients in the following three planes: sagittal (1025 images), axial (994 images), and coronal (1045 images) plane. Examples of different types of tumors, as well as different planes, are shown in Figure 20. The tumors are marked with a red outline. The number of images is different for each patient. Herein, each 3D volume includes 155 2D slices/images of brain MRIs collected at various locations across the brain. Every slice is 240 × 240 pixels in size in NIfTI format and is made up of single-channel grayscale pixels. With NIfTI files, images, and other data are stored in a 3D format. It is specifically designed this way to overcome the spatial orientation challenges of other medical image file formats. The results of the developed model are shown in Table 6 in terms of dice scores. The results reveal that the dice score can exceed 0.93 (on average) for different types of tumors in different MR1 scan planes, showing a good overlap with manual segmentations. The execution speed was quite good, with an average of less than 7 s per volume to test, i.e., 45 ms per image. Figure 21 confirms the ability of the suggested model to segment multiple tumor lumps per slice. Scholars recommend routinely taking axial images. In selected cases, coronal or sagittal planes may be added.

### Convergence Issue of the Proposed Model

Alignment, separation, cohesion, attraction to food sources, and distraction from enemy sources are the primary factors influencing the QDA algorithm’s exploration and exploitation. Dragonflies should adjust their weights in an adaptable manner as they go from exploitation to exploration. During the optimization phase, this ensures the convergence of dragonflies. The convergence of QDA was expected after a small number of iterations. However, the position updating rule of traditional QDA has a lower correlation with the centroid of the preceding generation’s population. As a consequence, this may cause problems in locating the global optimum, leading to a solution with poor precision and a tendency to converge too quickly to local minima [24,25,68,69]. Therefore, investigations are urged to discover new methods to update dragonfly locations. By utilizing the K-mean algorithm to determine the initial centroid of the generation’s population, the suggested method strengthened the algorithm’s capacity for exploration and led to a greater variety of solutions. In terms of convergence speed, the results shown in Table 7 proved that compared to the QDA without the k-mean algorithm, the proposed technique provided better performance with a 60% reduction in the number of required iterations for convergence to the optimal solution. Herein, Hausdorff 95 (95% HD) as an evaluation metric was used to measure the 95th percentile of the distances between boundary points in X (predicted segmentation results) and Y (ground truth). The purpose of using this metric is to eliminate the impact of a very small subset of the outliers.

## 5. Conclusions

This research proposes a reliable approach for extracting tumors from 3D MRI scans. The level set segmentation method, in its modified form, is used in the suggested model. In this updated version, clustering based on quantum-inspired DA is used to control the initial contour precisely rather than randomly through employing k-means for initial source food selection over randomly applied QDA. To make up for the shortcomings of the traditional DA-based clustering technique, such as delayed convergence or being stuck in a local optimum, the suggested model uses a quantum-inspired computing paradigm to stabilize the trade-off between exploitation and exploration. First, the quantum rotation gate concept may move a colony of agents to a location where they can better reach the optimal value. The mutation process enhances the swarm’s mutating and diversity, giving the primary approach substantial local search capability. QDA determines tumor shapes after disassembling the skull from the brain. These extracted edges will form the first MRI series contour. Finally, level set segmentation isolates the tumor across all volume segments. Results reveal that the proposed approach’s accuracy outperforms traditional dragonfly-based level set brain segmentation approaches by 2.5%. The limitations of the suggested model are that it focuses only on complete tumor segmentation and has not been tested on volumes including more than one tumor per slice. In order to further simplify the proposed model, future work may use the parallel segmentation technique. Additionally, the influence of variables (parameters) for level set segmentation and the dragonfly method on the model’s accuracy was investigated, and the proposed model was expanded to account for the various tumor components. Finally, the proposed approach may be employed in conjunction with a deep neural network to extract highly informative features for further segmentation.

## Figures and Tables

**Figure 1 bioengineering-10-00819-f001:**
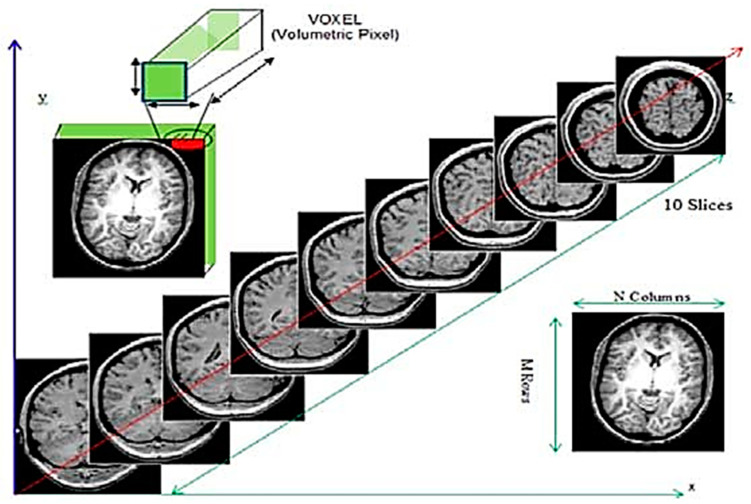
Voxel and slice in 3D MRI data. A slice is just like a 2D image stored in a matrix of size M × N. The smallest unit of a slice is a voxel i.e., a volumetric pixel with certain dimensions. MR data are a stack of 2D images acquired in 3D space while a person walking with a camera along any one of three spatial dimensions. If a person is lying on an MRI bed, the *z*-axis then becomes upward. The axial plane corresponds to XZ Plane, the Coronal plane corresponds to the XY plane and the Sagittal plane corresponds to the YZ plane.

**Figure 2 bioengineering-10-00819-f002:**
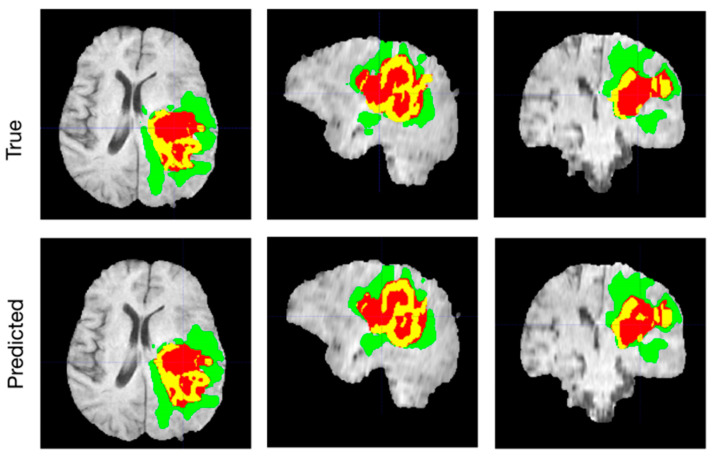
Automatically segmenting brain tumors. The whole tumor (WT) class includes all visible labels (a union of green, yellow, and red labels), the tumor core (TC) class is a union of red and yellow, and the enhancing tumor core (ET) class is shown in yellow (a hyperactive tumor part). The predicted segmentation results match the ground truth well.

**Figure 3 bioengineering-10-00819-f003:**
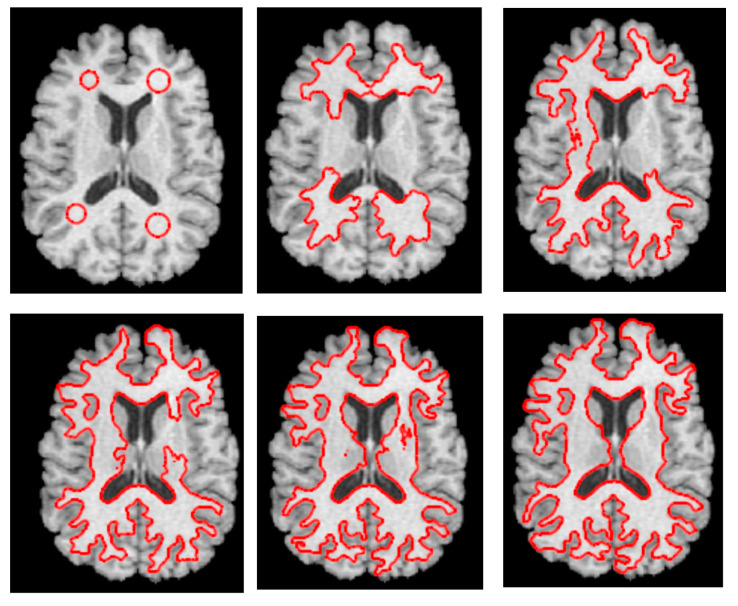
Demonstration of level set segmentation of white matter in a brain. An adaptive initial contouring method is performed to obtain an approximate circular contour of the tumor (red lines). Finally, the deformation-based level set segmentation automatically extracts the precise contours of tumors from each individual axial 2D MRI slice separately and independently. Temporal ordering is from left to right, top to bottom, to track the dynamic change of the contour of the tumor over different iterations (time).

**Figure 4 bioengineering-10-00819-f004:**
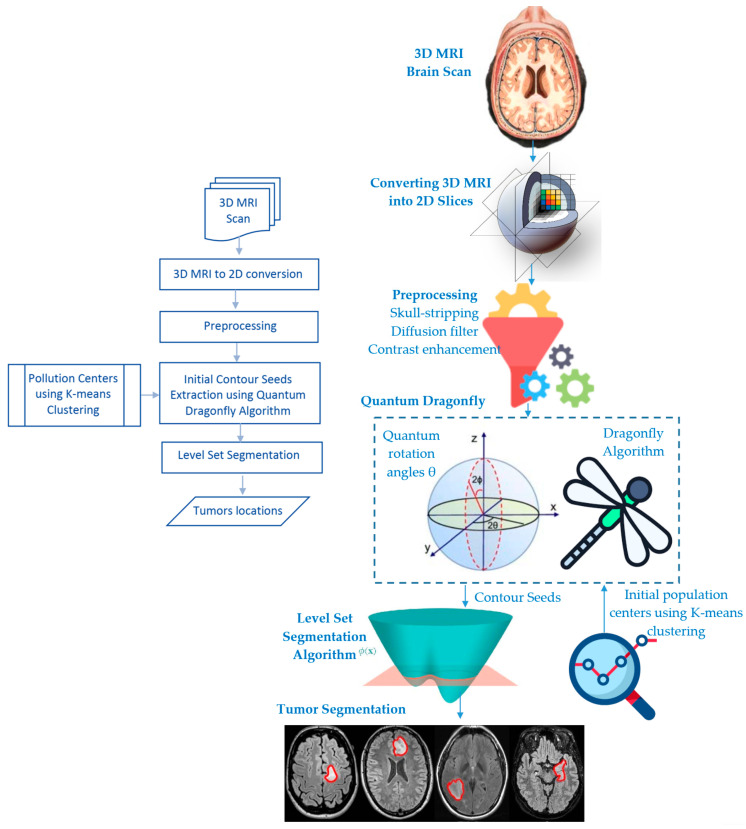
The suggested QDA-based methodology for detecting brain tumors: (**Left**) flowchart, (**Right**) graphical representation.

**Figure 5 bioengineering-10-00819-f005:**
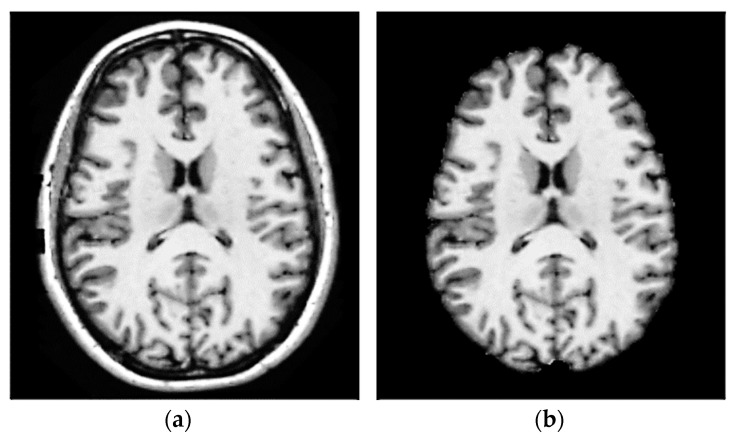
A consequence of skull-stripping MRI on the brain. (**a**) Tissue from the initial MRI image of the brain, and (**b**) brain without the skull.

**Figure 6 bioengineering-10-00819-f006:**
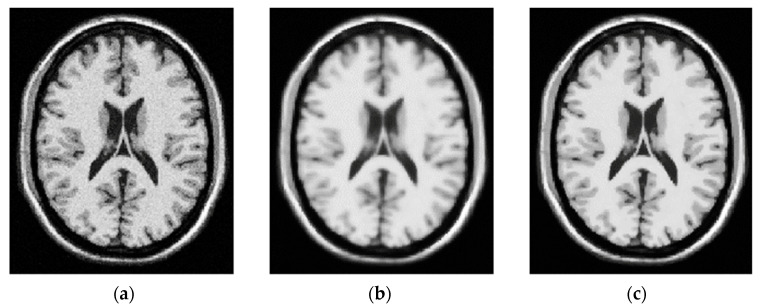
(**a**) Synthetic MR brain image, axial section, maximum intensity noise (5%); (**b**) image filtered with fixed Gaussian window size; (**c**) image filtered with decreasing window size at the same number of iterations. A Gaussian Filter is a low-pass filter used for reducing noise (high-frequency components). The kernel is not hard on drastic color changes (edges) due to the pixels towards the center of the kernel having more weightage towards the final value than the periphery.

**Figure 7 bioengineering-10-00819-f007:**
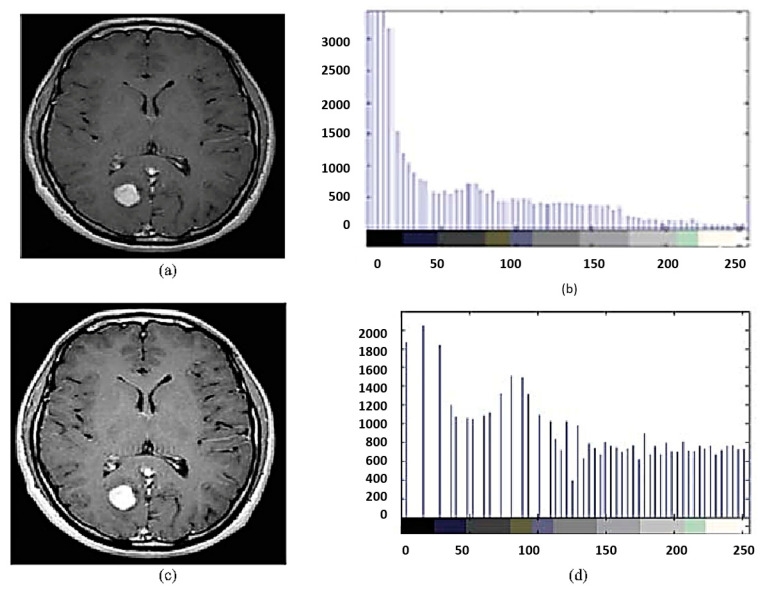
Histogram equalization technique (**a**) original image, (**b**) histogram image for (**a**), (**c**) histogram equalized image for (**a**), (**d**) histogram image for (**c**) equalize the two sub images.

**Figure 8 bioengineering-10-00819-f008:**
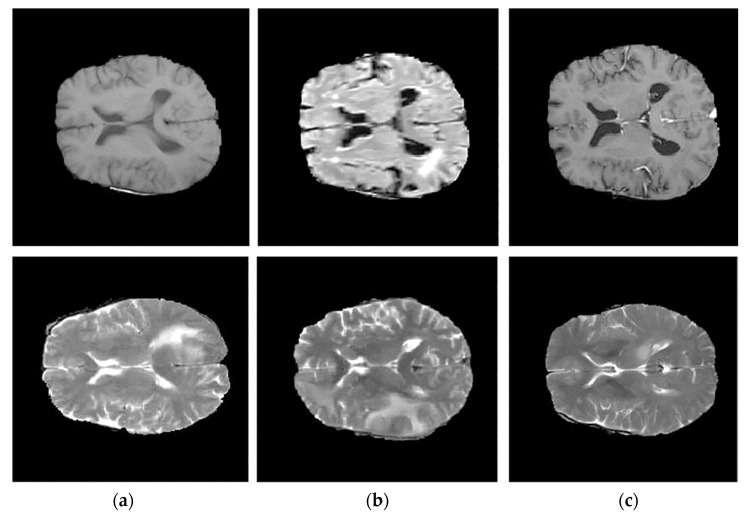
Removal of noise in MRI images (**a**) normal MRI, (**b**) noisy MRI, (**c**) denoised MRI for both T1 modality (Upper row) and T2 modality (Lower row). The role of sigma in the Gaussian filter is to control the variation around its mean value. So as the Sigma becomes larger the more variance allowed around mean and as the Sigma becomes smaller the less variance allowed around mean.

**Figure 9 bioengineering-10-00819-f009:**
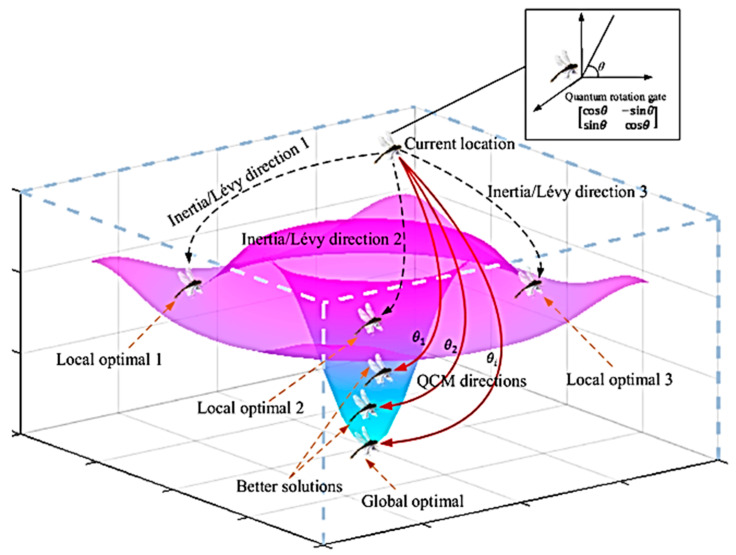
QDA with various quantum rotation angles *θ*, to reach to best solutions eventually, to global optimal. During the searching journey, a dragonfly individual has several directions to move to locally optimal solutions (Local optimal 1, 2, and 3) based on their inertia search directions or Lévy flight limitations; QDA is utilized to replace these two searching behaviors and escape from the local solution.

**Figure 10 bioengineering-10-00819-f010:**
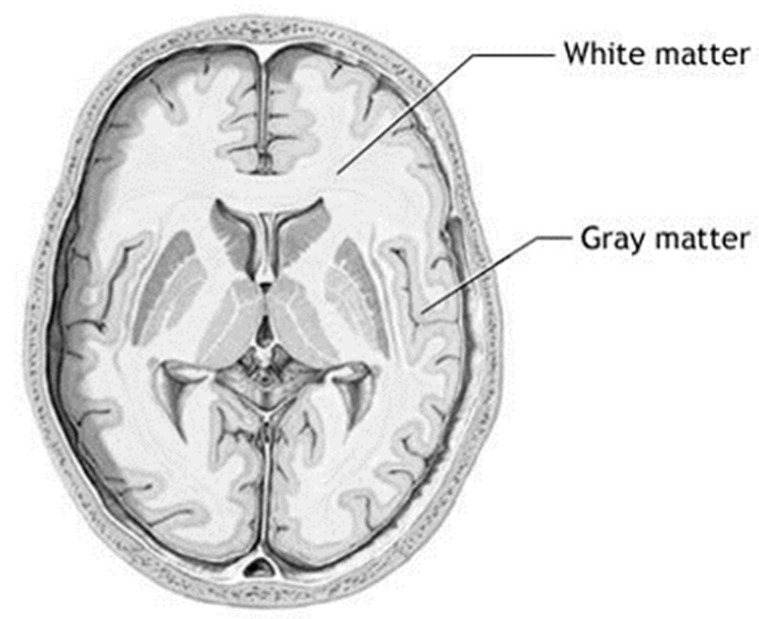
Brain axial section gray matter and white matter. Micrograph showing normal white matter (left of image-lighter shade of pink) and normal grey matter (right of image-dark shade of pink). Gray matter is made up of neuronal cell bodies, while white matter primarily consists of myelinated axons. In the brain, white matter is found closer to the center of the brain, whereas the outer cortex is mainly grey matter.

**Figure 11 bioengineering-10-00819-f011:**
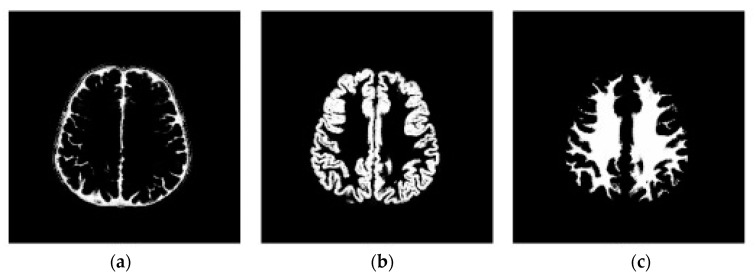
(**a**) Segmented cerebrospinal fluid (CSF), (**b**) segmented gray matter, and (**c**) segmented white matter. The three-dimensional MRI T1 brain image was considered with the following five layers: scalp, skull, cerebral spinal fluid (CSF), gray matter, and white matter.

**Figure 12 bioengineering-10-00819-f012:**
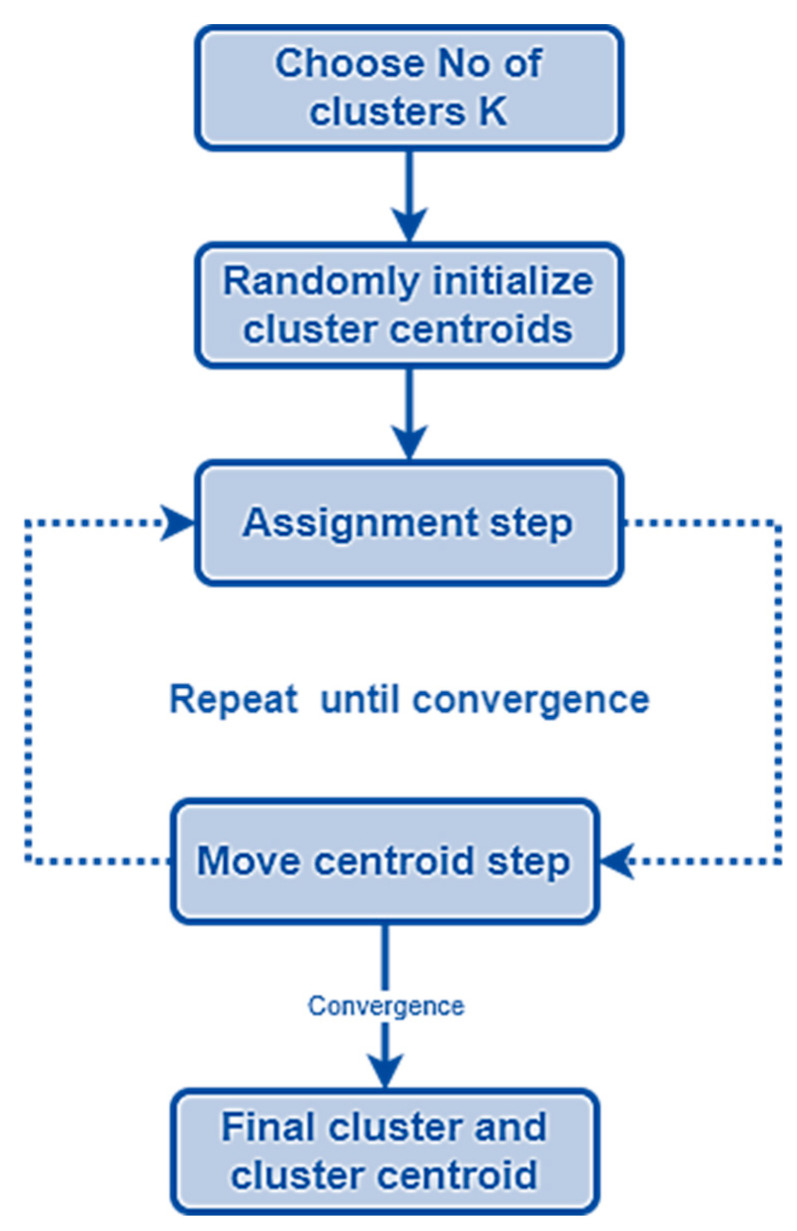
K-means clustering flowchart. The k-means method aims to divide a set of *N* objects into *k* clusters, where each cluster is represented by the mean value of its objects.

**Figure 13 bioengineering-10-00819-f013:**
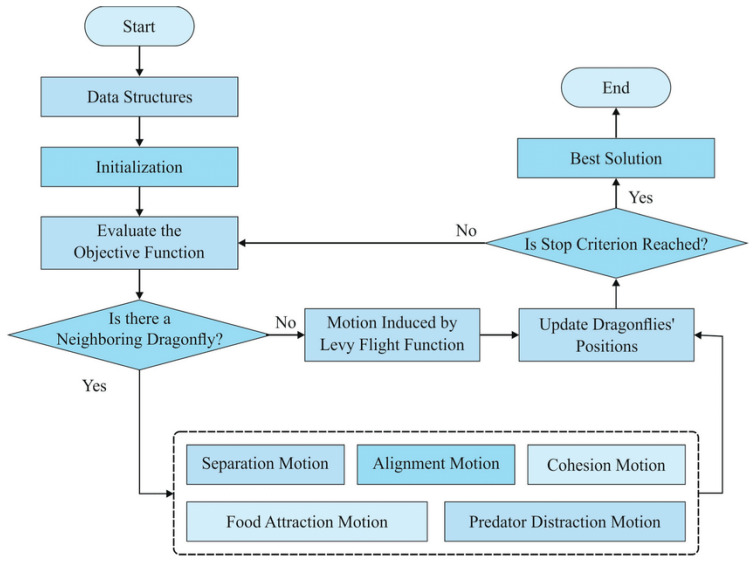
Flowchart of Dragonfly algorithm. Dragonflies form sub-swarms and fly over different areas in a static swarm. This is similar to exploration, and it aids the algorithm in locating appropriate search space locations. On the other hand, dragonflies in a dynamic swarm fly in a larger swarm and in the same direction. In addition, this type of swarming is the same as using an algorithm to assist it converges to the global best solution.

**Figure 14 bioengineering-10-00819-f014:**
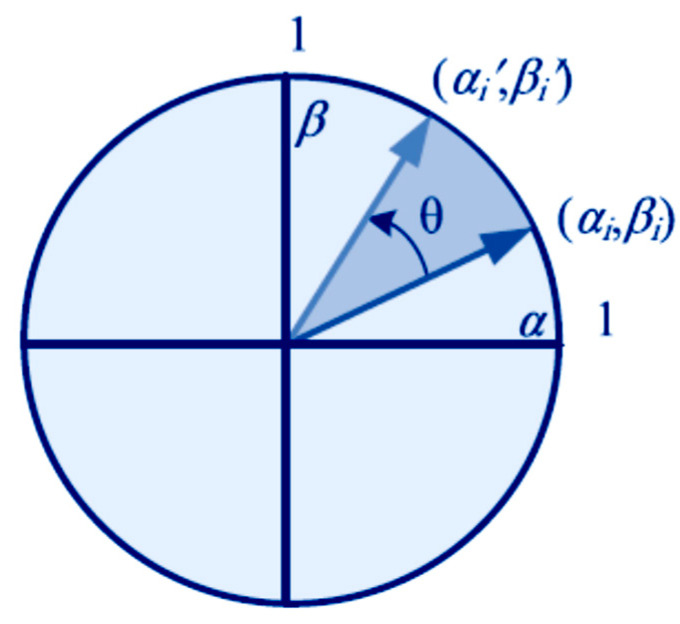
The updating of a quantum bit state vector, where ${\left(\alpha_{i},\beta_{i}\right)}^{T}$ and ${\left({\overset{´}{\alpha }}_{i},{\overset{´}{\beta }}_{i}\right)}^{T}$ show the quantum bit state vector before and after the rotation gate updating of the *i*th quantum bit of chromosome; $\theta_{i}$ shows the *i*th rotation angle to control the convergence rate. The update strategy of the quantum chromosome in the quantum rotation gate is to compare the fitness of the current individual with that of the optimal individual, select the better one, and then rotate to it.

**Figure 15 bioengineering-10-00819-f015:**
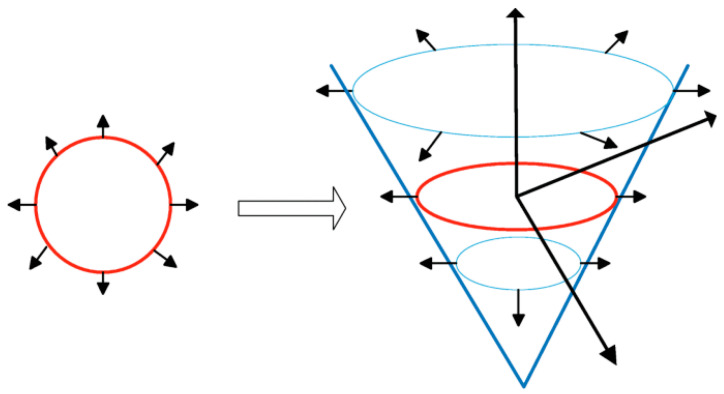
Level set function—An overview. The level set approach takes the original curve (the red one on the left) and builds it into a surface. That cone-shaped surface, which is shown in blue on the right below, has a great property; it intersects the XY plane exactly where the curve sits. The blue surface is called the level set function because it accepts as input any point in the plane and hands back its height as output. The red front is called the zero level set because it is the collection of all points that are at height zero.

**Figure 16 bioengineering-10-00819-f016:**
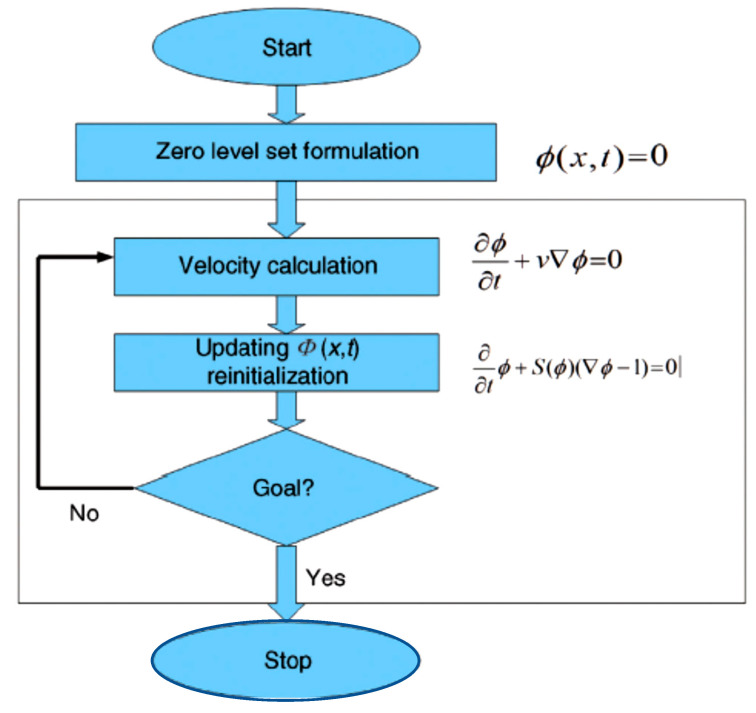
General image segmentation algorithm using the level set function. Given a certain area Ω with an edge Γ. The velocity of the edge *ν* between steps depends on the position, shape, time, and external conditions. The function $\varphi \left(x,t\right)$ where *x* is the position in the Cartesian space and *t* is the time, describing the moving contour. $E\left(\varphi \right)$ is the energy, $R_{p}\left(\varphi \right)$ is the level set regularization term, $L_{p}\left(\varphi \right)$ is minimized when the zero level contour is located at the object boundaries and $S_{g}\left(\varphi \right)$ is introduced to speed up the motion of the zero level contour in the level set evolution process.

**Figure 17 bioengineering-10-00819-f017:**
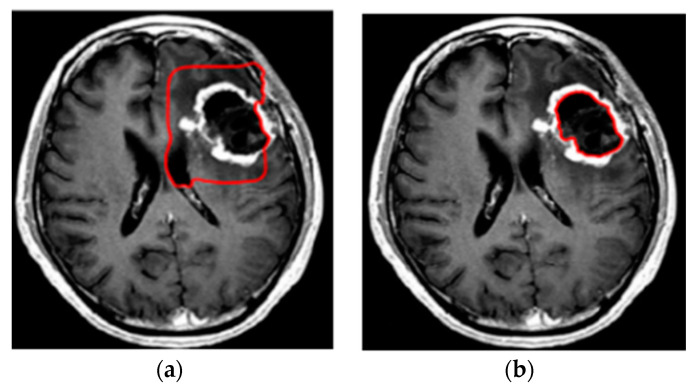
Example of level set brain tumor segmentation. (**a**) Original image with initial contours (red line) based on a clustering method influenced by QDA to accurately extract initial contour points (**b**) segmented tumor using level set function.

**Figure 18 bioengineering-10-00819-f018:**
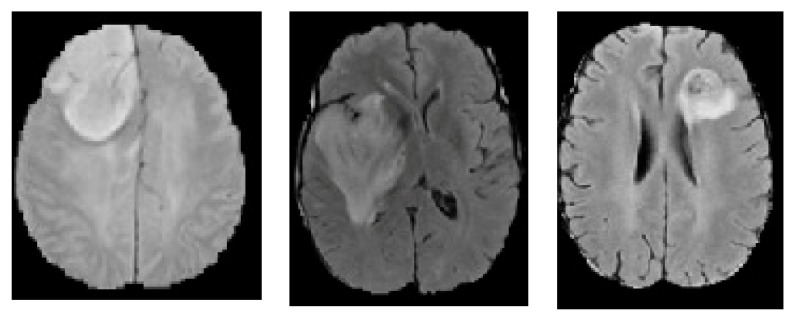
Brain tumor segmentation (First row) 2D slice (Second row) Final segmentation using QDA (blue areas).

**Figure 19 bioengineering-10-00819-f019:**
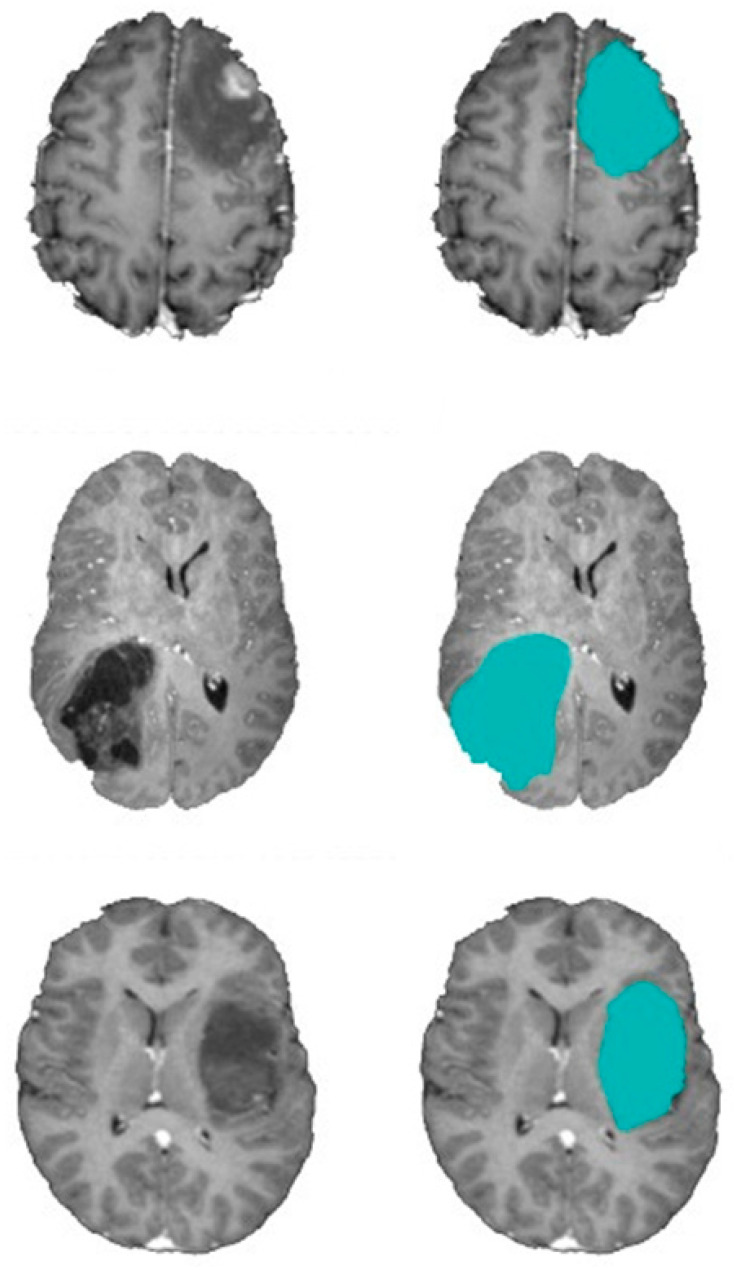
Brain tumor segmentation (left column) 2D slice (right column) Final segmentation using QDA (blue areas).

**Figure 20 bioengineering-10-00819-f020:**
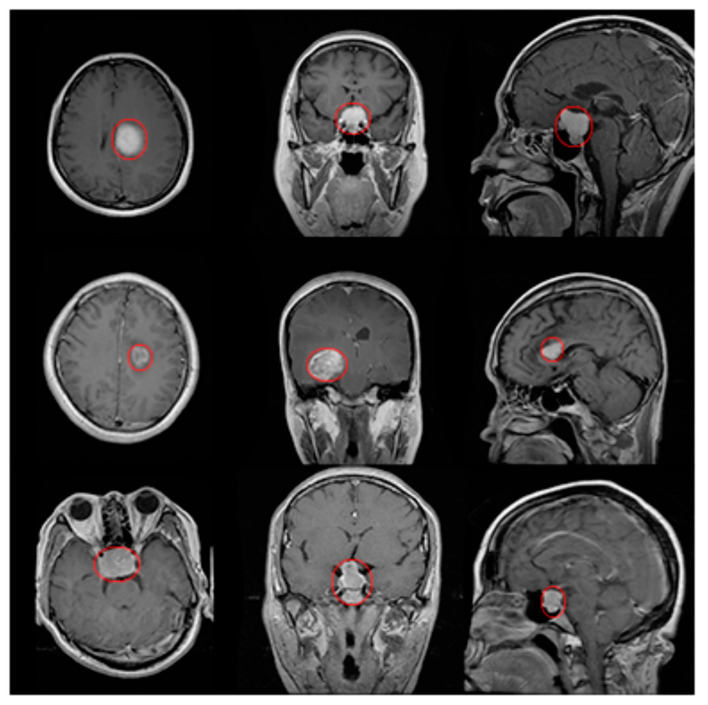
MRI scans of different tumor types in different planes—the red circles highlight the tumor in the images. The example is shown for each tumor type in each plane. The first row is for the meningioma, which is a tumor that arises from the meninges. The second row is for glioma, which is the growth of cells. That third row is for pituitary tumors, which are unusual growths that develop in the pituitary gland. The columns from left to right represent the MRI image scans from axial, coronal, and sagittal planes.

**Figure 21 bioengineering-10-00819-f021:**
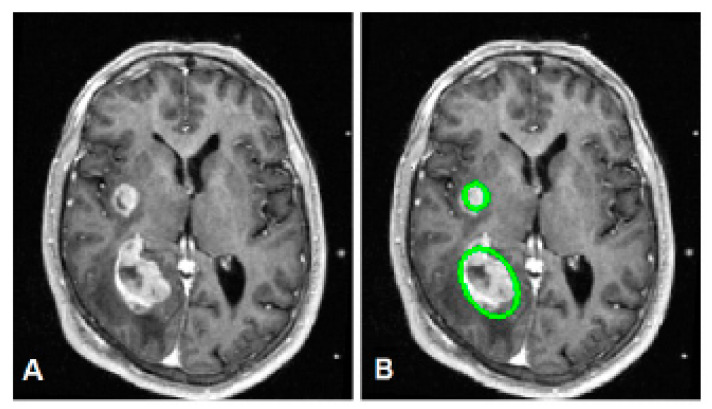
(**A**) Original image, showing two tumors in a representative axial slice; (**B**) the detection result of our proposed method (green circles).

**Table 1 bioengineering-10-00819-t001:** Comparison between level set-based MRI brain tumor segmentation approaches.

	Algorithm-Based	Advantages	Disadvantages
Shu, X. et al. [30]	Level set with split Bregman technique	3D segmentation model	Low accuracy when dealing with heterogeneous tumors
Yang, Y. et al. [31]	Two-level set segmentation based on mutual exclusion	Ensure the independence of neighboring regions	Require prior knowledge for parameter initialization
Lei, X. et al. [10]	Sparse constrained level set segmentation	Accurate to identify common features of the shape of brain tumors	Inaccurate segmentation results when dealing with noisy images
Song, J., Zhang, Z. [32]	Weighted level set model	Segmenting MR images with inhomogeneous intensity	Not segment 3D MRI images directly
Jin, R. et al. [33]	Level set with a constraint term	High accuracy when the number of tissues and the level of noise grow	The setting of the optimal threshold is very subjective
Khosravanian, A. et al. [34]	Fuzzy shape prior term with deep learning	Handling contour leakage and shrinkage	Need complex network architecture
Shu, X. et al. [35]	Adaptive local variance-based level set	Accurate and noise resilience	Require prior knowledge for parameter initialization
Kalam, R. et al. [36]	Modified region growing with neuro-fuzzy classifier	Segmenting MR images with inhomogeneous intensity	Require prior knowledge for parameter initialization and the post-processing step
Pang, Z. et al. [37]	Adaptive weighted curvature, with heat kernel convolution	Reduce the computation complexity	The setting of the optimal threshold is very subjective
Dhamija, T. et al. [38]	Level set with two deep learning models	Handling different segmentation tasks	Require high computational resources
Zhuang, M. et al. [39]	Iterative deep learning technique with boundary representation	Enhance the precision of boundary identification	Need complex network architecture
Khan, M., et al. [40]	Active contour and deep learning feature optimization	Segmenting MR images with inhomogeneous intensity	Need complex network architecture

**Table 2 bioengineering-10-00819-t002:** Analysis of the proposed model vs. related level set-based brain segmentation techniques (average across 285 participants in the BraTS 2019 dataset).

	Accuracy (%)	Recall (%)	Precision (%)	Dice Score	Specificity
Symmetry Analysis, Level Set [85]	93.43	89.15	90.60	0.911	0.931
K-means, Level Set [86]	89.45	92.87	75.97	0.902	0.945
ANN, Level Set [87]	96.80	95.30	94.16	0.940	0.923
Local edge features, Weighted level set [88]	95.62	94.19	93.57	0.920	0.965
Dragonfly, Level Set [59]	96.21	95.15	93.85	0.923	0.987
Proposed Model	98.95	97.36	95.14	0.947	0.993

**Table 3 bioengineering-10-00819-t003:** Analysis of the proposed model vs. DNN-based brain segmentation techniques (average across 285 participants in the BraTS 2019 dataset).

Models	Accuracy (%)	Recall (%)	Precision (%)	Dice Score	Specificity
DCNN, level set [89]	93.43	89.15	90.60	0.913	0.991
DCNN, symmetric mask [90]	89.45	92.87	75.97	0.828	0.891
DCNN, SVM [87]	96.80	95.30	94.36	0.932	0.989
Proposed Model	98.95	97.36	95.14	0.948	0.995

**Table 4 bioengineering-10-00819-t004:** Analysis of the proposed model vs. quantum-inspired metaheuristic brain segmentation techniques (average across 285 participants in the BraTS 2019 dataset).

Models	Accuracy (%)	Recall (%)	Precision (%)	Dice Score	Specificity
QDA, Level Set, mutation procedure (Proposed)	98.97	97.40	95.20	0.947	0.992
PSO, Level Set [92]	97.30	95.13	94.90	0.941	0.987
ABC, Level Set [93]	97.68	95.67	94.86	0.945	0.989
CF, Level Set [94]	97.87	95.95	94.35	0.939	0.990

**Table 5 bioengineering-10-00819-t005:** Analysis of the proposed model accuracy with and without k-means (average over 285 participants in the BraTS 2019 dataset).

Models	Accuracy (%)	Mean	Standard Deviation	Median	25 Quartile	75 Quartile
k-means, QDA, and level set	98.95	0.976	0.032	0.975	0.972	0.979
QDA, and level set	87.63	0.845	0.064	0.839	0.837	0.840

**Table 6 bioengineering-10-00819-t006:** Analysis the efficiency of the suggested model in terms of dice scores to segment different types of tumors in different planes.

Tumor Type/Plane	Axial	Coronal	Sagittal
Meningioma	0.913	0.923	0.985
Glioma	0.939	0.940	0.986
Pituitary	0.947	0.943	0.909

**Table 7 bioengineering-10-00819-t007:** Analysis of the proposed model convergence speed with and without k-means (average over 285 participants in the BraTS 2019 dataset).

Models	Hausdorff 95	Convergence Speed
k-means, QDA, and level set	4.41	30 independent runs
QDA, and level set	6.92	75 independent runs

## Data Availability

Datasets for this research are available in https://www.kaggle.com/datasets/aryashah2k/brain-tumor-segmentation-brats-2019/code?resource=download (accessed on 1 October 2022).

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
