# Peer review of "A New Medical Analytical Framework for Automated Detection of MRI Brain Tumor Using Evolutionary Quantum Inspired Level Set Technique"

_bioengineering, 2023, doi:10.3390/bioengineering10070819_

Round 1
Reviewer 1 Report
1. Please summarize most important references in Table form by providing their advantages and disadvantages etc. This must be up-to-date.
2. Under Section 3: The Proposed Evolutionary Quantum Brain Tumor Detection Method, please summarize the algorithm in flow chart diagram.
3. Please provide detail explanation on where is the novelty of the proposed methods compared with existing methods.
4. Instead of DragonFly, how does the other methods can be implemented under the same proposed framework? Please add the numerical results and simulation on this.
5. Since the authors only using Accuracy, Recall and Precision as the main measurements, there might be the main results are not comprehensive because we have many other measurement. Please look at into this matter.
6. Add more statistical-goodness fit measurements to validate the model.
7. Discuss also the convergence issue of the proposed model.
8. Some images are in the gray scale mode. Some got color mode. Please clarify on this.
9. Once again, the methodology as well as the computer implementation need further detail explanation.
can be further improved.
Author Response
To The Editor-in-chief and the Associate Editor,
Bioengineering.
|
Title: A New Medical Analytical Framework for Automated Detection of MRI Brain Tumor using Evolutionary Quantum Inspired Level Set Technique
Dear Sir,
Thanks for giving us the opportunity to revise the paper for the possible publication in your interesting journal. We also thank the reviewers for their constructive comments. We have significantly revised the manuscripts based the comments received and the details are summarized below.
Reply to the comments:
In the revised version, what was modified and added was written in red
Reviewer 1
|
S. No |
Comments |
Changes made in the revised manuscript based on the comments |
|
1 |
Please summarize most important references in Table form by providing their advantages and disadvantages etc. This must be up-to-date. |
All papers mentioned in the related work section are recent in the years 2021–2023.
In the revised version, a new table was added that summarizes the related papers mentioned in the related work section by providing their advantages and disadvantages. (Table 1-Page 8) |
|
2 |
Under Section 3: The Proposed Evolutionary Quantum Brain Tumor Detection Method, please summarize the algorithm in flow chart diagram. |
Figure 4 shows the data flow diagram of the proposed method in a graphical representation style
In the revised version, the flowchart diagram of the suggested framework was added to Figure 4. (Page 10)
|
|
3 |
Please provide detail explanation on where is the novelty of the proposed methods compared with existing methods. |
In the revised version, a new paragraph was added in subsection 1.2 to clarify the novelty of the suggested framework.
“In this study, for optimization in the selection of the initial contour points, dragonfly behaviors are quantimized using quantum computing to improve the DA's search efficiency. To go from a local optimal solution to a better solution (escape from the local solution) and hopefully to the global optimum, each dragonfly may employ a quantum rotation gate to overcome the inertia weight during the search operations and to conquer the searching quality of Levy flight”. (Page 5)
Furthermore, a new paragraph was added to subsection 2.1 to illustrate the novelty of the suggested framework as different from the current methods.
“The development of a novel evolutionary quantum-inspired level set segmentation approach for a brain tumor detection system based on a 3D-MRI has, however, received little attention, as far as we are aware. The potential benefits of quantum ma-chine learning include increased speed and accuracy, improved scalability, and a more efficient use of resources. Additionally, quantum algorithms can provide insights into data that would be difficult to uncover with classical algorithms [41] [42]. (Page 9) |
|
4 |
Instead of Dragonfly, how does the other methods can be implemented under the same proposed framework? Please add the numerical results and simulation on this. |
In the revised version, a new set of experiments was conducted to investigate how other metaheuristics methods will be employed instead of dragonfly under the same proposed model.
“Despite the fact that quantum clustering and the level set technique are not particularly novel to the area of brain tumor segmentation, an additional series of experiments was done to validate their effectiveness when combined. The suggested model's QDA component has been swapped out with a set of well-known quantum –inspired metaheuristic modules running in black box mode with their initial settings preserved. Quantum Particle swarm optimization (QPSO), Quantum artificial bee colony (QABC) algorithm, and Quantum clown fish (QCF) method are all examples of employed quantum-inspired metaheuristic algorithms [92-94]. The results shown in Table 4 support the study's hypothesized improvement in segmentation accuracy when utilizing the QDA classifier based on correct center points (initial contour seeds) retrieved using k-means. When compared to the closest combination between the QCF and the level set segmentation, the proposed combination improved accuracy by at least 1%. Whether a QPSO, QABC, or QCF algorithm is more exploratory or exploitative de-pends on the convergence rate, which is in turn determined by the major factors that affect the individual's movement towards the best position found so far. The convergence rate in QDA is parameter-free. The suggested model has a robust local search capacity by adopting a mutation procedure to enhance the swarm's mutating and realizing its variety”.
Table4 - Page 23 |
|
5 |
Since the authors only using Accuracy, Recall and Precision as the main measurements, there might be the main results are not comprehensive because we have many other measurement. Please look at into this matter. |
In the revised version, other evaluation metrics such as dice score, specificity, and Hausdorff 95 were added to the results in Tables 2, 3, 4, and 6 to make the obtained results more comprehensive. Pages 21, 23, and 25 |
|
6 |
Add more statistical-goodness fit measurements to validate the model. |
In the revised version, more statistical goodness fit measurements such as Median, 25th Quartile, and 75th Quartile were used to validate the suggested model. Table 5- page 24 |
|
7 |
Discuss also the convergence issue of the proposed model. |
In the revised version, a new subsection 5.1 “Convergence issue of the proposed model” was added to discuss the convergence speed of the suggested model.
“Alignment, separation, cohesion, attraction to food sources, and distraction from enemy sources are the primary factors influencing the QDA algorithm's exploration and exploitation. Dragonflies should adjust their weights in an adaptable manner as they go from exploitation to exploration. During the optimization phase, this ensures the convergence of dragonflies. The convergence of QDA was expected after a small number of iterations. However, the position updating rule of traditional QDA has a lower correlation with the centroid of the preceding generation's population. As a consequence, this may cause problems in locating the global optimum, leading to a so-lution with poor precision and a tendency to converge too quickly to local minima [24][25][68][69]. Therefore, investigations are urged to discover new methods to up-date dragonfly locations. By utilizing the K-mean algorithm to determine the initial centroid of the generation's population, the suggested method strengthened the algorithm's capacity for exploration and led to a greater variety of solutions. In terms of convergence speed, the results shown in Table 6 proved that compared to the QDA without the k-mean algorithm, the proposed technique provided better performance with a 60% reduction in the number of required iterations for convergence to the optimal solution. Herein, Hausdorff 95 (95% HD) as an evaluation metric was used to measure the 95th percentile of the distances between boundary points in X (predicted segmentation results) and Y (ground truth). The purpose of using this metric is to eliminate the impact of a very small subset of the outliers”
Table 6- pages 24 and 25 |
|
8 |
Some images are in the gray scale mode. Some got color mode. Please clarify on this. |
In the revised version, the brain tumor images are unified to be in grayscale , but all automatically segmenting brain tumors were colored to highlight the contour of tumors and tumor regions.
All other Figures (non-brain images) were colored to view their content more obviously. |
|
9 |
Once again, the methodology as well as the computer implementation need further detail explanation. |
In the revised version, many sentences were added to clarify the methodology as well as the computer implementation.
“Open-source software platform 3D Slicer (https://www.slicer.org) is utilized for the conversion. It is used for medical image informatics, image processing, and 3D visualization”.
“Three morphological procedures are performed during the skull-stripping process: Otsu's thresholding is used to transform the input 2D-MRI slices to binary images in the first step. Step 2 involves creating a mask from 2D MRI scans of the brain by diluting and eroding the slices. The brain becomes a connected, complete component by filling up the holes. As a third superimposed step: The final skull-strip image is generated by superimposing the mask over the original image”
“Noise may be reduced in images using a technique called anisotropic diffusion, which preserves important details like edges and lines without distorting the rest of the image”
“In anisotropic diffusion, the original image is combined with a filter that is itself dependent on the local content of the original image to get a family of parameterized images”
“In the final step, contrast enhancement is applied by making use of the histogram equalization technique. This technique improves the visibility of tumors by rearranging the grayscale of the images in a non-linear fashion.”
“This study quantamize dragonfly behaviors using a quantum computing technique, creating a new algorithm called Quantum Dragonfly (QDA) that improves the drag-onfly algorithm's search efficiency, as illustrated in Figure 9 [21] [22]. Since the QDA has never been hybridized with the clustering model, this study focuses on combining the DA and the QCM with a clustering model in order to address its limitations”
“In our model, the AD technique for clustering issues is enhanced by applying the k-means algorithm, and quantum computing to speed the convergence rate while preserving the balance between exploration and exploitation. Here, K-means is utilized to construct the correct centers to deal with the randomization to determine the center of mass of the neighborhood inside the traditional DA algorithm.”
Figure 12 shows the main steps of the k-mean clustering procedure
Figure 13 illustrates the flowchart of Dragonfly algorithm
“Partial differential equations (PDE), which involve progressively evaluating the differences between neighboring pixels to determine object boundaries, provide the ba-sis of a large class of contemporary image segmentation algorithms, among which lev-el sets are an important subclass [74].”
Fig. 16 illustrates the level-set steps in medical image segmentation.
The experiment was run in MATLAB R2018a on a computer with an Intel (R), Core (TM) i3 CPU and 8.00 GB of RAM.
|
Reviewer 2 Report
It looks like the authors have taken some well-known numerical and image processing techniques established elsewhere, combined, modified and applied them to a worthy situation (brain tumour identification). I haven’t been closely following use of segmentation techniques in bio or medical fields – not really my fields – so I’m not able to judge if the paper contains enough distinctive novelties in those fields. But it’s certainly an interesting application and I’m inclined to accept their claims at face value.
Figure captions in most cases are too simplistic. They should aim to be relatively self-contained, i.e., without consulting the main text, and just from the caption the reader can understand what the figure is showing and what point(s) it’s helping to make.
Figure 2: unclear what the 3 colours signify. Not explained in the main text, should be mentioned in caption.
Level-set is usually used to track dynamic change of the contour of something over time. On first sight of Figure 3 and for a brief moment, I thought they were using level-set to follow progress of some feature of interest through the slices to get a 3D shape of the tumour. That would have been a novel thing to do and a faith leap (to apply something meant for a temporal sequence over time to a spatial sequence over distance). Then as I read the caption I realised I misunderstood the figure. I think level-set was used here to detect tumour in each individual slice separately and independently.
Just a thought: in the paper, as in most applications, axial slices were used. What if the techniques are applied to coronal and sagittal slices as well. Some differences in results are expected. Would the combined version be more useful/accurate?
It’s possible to use level-set directly on a 3D shape. A follow up paper using 3D level-set to deal with the volume, rather than slice by slice, would be of interest.
It’s unclear to me but I think it’s important to clarify, at conceptual level, what you’re relying on to decide if a pixel belongs to a tumour or not. Pixel values, or pixel value deviation from the norm, pixel grouping according to some criteria, or something else? Lines 403-406 appear to suggest that pixel value is the key. If so, do you really need to use all those sophisticated techniques to identify the tumour pixels, particularly if you’re only
Figure 6: wouldn’t Gaussian filtering change/hide details of features of interest?
Figure 8: not my field so I don’t really know. But I’m just wondering, the original images would undergo a lot of changes to make them segmentation friendly. Can one still trust results from the somewhat “doctored” images?
In Section 5, memory usage was indicated. Be nice to also let the reader to have a feel about the runtime (seconds, mins, hours or days)?
In conclusions, after all the troubles, improvement as stated was 2.5%. Comment on significance of this figure should help the reader to better appreciate the effort and importance. Also state the baseline, i.e., 2.5% on top of what value, 96.2%? Are we chasing a diminishing return here? The method hasn’t been tested for multiple tumour lumps per slice. This is a serious limitation, isn’t it? Because a single cancerous mass may show up as separate lumps in a slice. Also what does it mean in practice? Someone human needs to make a judgement that there is only one tumour to look for before they feed the data to your model to quantify it?
Author Response
To The Editor-in-chief and the Associate Editor,
Bioengineering.
|
Title: A New Medical Analytical Framework for Automated Detection of MRI Brain Tumor using Evolutionary Quantum Inspired Level Set Technique
Dear Sir,
Thanks for giving us the opportunity to revise the paper for the possible publication in your interesting journal. We also thank the reviewers for their constructive comments. We have significantly revised the manuscripts based the comments received and the details are summarized below.
Reply to the comments:
In the revised version, what was modified and added was written in red
Reviewer 2
|
S. No |
Comments |
Changes made in the revised manuscript based on the comments |
|
1 |
But it’s certainly an interesting application and I’m inclined to accept their claims at face value. |
Thank you for the positive feedback. |
|
2 |
Figure captions in most cases are too simplistic. They should aim to be relatively self-contained, i.e., without consulting the main text, and just from the caption the reader can understand what the figure is showing and what point(s) it’s helping to make.
Figure 2: unclear what the 3 colors signify. Not explained in the main text, should be mentioned in caption. |
In the revised version, all figure captions were modified to be relatively self-contained and make it easy for the reader to understand what the figure is showing and what point(s) it’s helping to make. Examples are
“Figure 1: Voxel and slice in 3D MRI data. A slice is just like a 2D image stored in matrix of size M × N. The smallest unit of slice is voxel i.e. volumetric pixel with certain dimensions. MR data is as a stack of 2D images acquired in 3D space while a person walking with camera along any one of three spatial dimensions. If a person is lying on MRI bed, z-axis then becomes upward. Axial plane corresponds to XZ Plane, Coronal plane corresponds to XY plane and Sagittal plane corresponds to YZ plane”.
“Figure 2: Automatically segmenting brain tumors. The whole tumor (WT) class includes all visible labels (a union of green, yellow and red labels), the tumor core (TC) class is a union of red and yellow, and the enhancing tumor core (ET) class is shown in yellow (a hyperactive tumor part). The predicted segmentation results match the ground truth well”. |
|
3 |
Level-set is usually used to track dynamic change of the contour of something over time. On first sight of Figure 3 and for a brief moment, I thought they were using level-set to follow progress of some feature of interest through the slices to get a 3D shape of the tumor. That would have been a novel thing to do and a faith leap (to apply something meant for a temporal sequence over time to a spatial sequence over distance). Then as I read the caption I realized I misunderstood the figure. I think level-set was used here to detect tumor in each individual slice separately and independently. |
Ok, in the revised version, and to remove this confusion, the Figure 3 caption was modified to be:
Figure 3: Demonstration of level-set segmentation of white matter in a brain. An adaptive initial contouring method is performed to obtain an approximate circular contour of the tumor. Finally, the deformation-based level set segmentation automatically extracts the precise contours of tumors from each individual axial 2D MRI slice separately and in-dependently. Temporal ordering is from left to right, top to bottom, to track the dynamic change of the contour of the tumor over different iterations (time). |
|
4 |
Just a thought: in the paper, as in most applications, axial slices were used. What if the techniques are applied to coronal and sagittal slices as well? Some differences in results are expected. Would the combined version be more useful/accurate? |
In the revised version, a new set of experiments was conducted to validate the efficiency of the proposed model when dealing with other types of MRI image scans, such as coronal and sagittal slices.
“To confirm the efficiency of the suggested model to segment different types of tumors with multiple tumor lumps per slice in different planes, the last set of experiments was conducted using a benchmark brain tumor dataset from the website https://figshare.com/articles/dataset/brain_tumor_dataset/1512427/5. In this dataset, there are three types of tumors: meningioma (708 images), glioma (1426 images), and pituitary tumor (930 images). All images were acquired from 233 patients in three planes: sagittal (1025 images), axial (994 images), and coronal (1045 images) plane. Examples of different types of tumors, as well as different planes, are shown in Figure 19. The tumors are marked with a red outline. The number of images is different for each patient. Herein, each 3D volume includes 155 2D slices/images of brain MRIs col-lected at various locations across the brain. Every slice is 240×240 pixels in size in NIf-TI format and is made up of single-channel grayscale pixels. With NIfTI files, images and other data are stored in a 3D format. It's specifically designed this way to over-come the spatial orientation challenges of other medical image file formats. The results of the developed model are shown in Table 6 in terms of dice scores. The results reveal that the dice score can exceed 0.93 (on average) for different types of tumor in different MR1 scan planes, showing good overlap with manual segmentations. The execution speed was quite good, with an average of less than 7 seconds per volume to test, i.e., 45 ms per image. Figure 20 confirms the ability of the suggested model to segment multiple tumor lumps per slice. Scholars recommend routinely taking axial images. In selected cases, coronal or sagittal planes may be added”.
Figure 19- Page 27 |
|
5 |
It’s possible to use level-set directly on a 3D shape. A follow up paper using 3D level-set to deal with the volume, rather than slice by slice, would be of interest. |
Yes, there are many papers that use level-set directly to segment 3D shapes that deal with volume rather than slice by slice. However, their usefulness has been limited by two problems. First, 3D-level sets are relatively slow to compute. Second, their formulation usually entails several free parameters, which can be very difficult to correctly tune for specific applications.
In the revised version, new sentences were added to clarify why 3D-MRI images are flattened into 2D slices before segmentation, as is followed in the majority of research that deals with this topic.
“Direct automated segmentation of objects in 3D medical imaging is a challenging task since it often requires effectively identifying a large number of separate structures with complicated geometries inside a large volume under examination. Surface determination, where the boundary between one area and another is precisely captured, is a crucial notion in 3D image segmentation. The scan's grayscale data is utilized to pin-point precisely where these boundaries are. The exact procedure is highly variable, de-pending on 3D image type and quality.
In response to these difficulties, the majority of modern machine learning strategies (e.g., deep learning) boost their learning potential by including more trainable parameters in their models. Since clinical imaging systems typically have low-end computer hardware with limited memory, CPU resources only, and a long inference time, in-creased model complexity will incur high computational costs and large memory requirements, making them unsuitable for real-time implementation on standard clinical workstations [1][2][6]. Consequently, due to the remote procedures, high-performance computing hardware such as high-end Graphics Processing Units (GPUs) is typically required on remote servers to provide the large memory requirements for model training and to accelerate inference speed by segmenting MR examinations through parallel computation, which, because of the distant nature of the operations, are not easily accessible for use in real time on clinical workstations. Moreover, trade-off techniques such as patch-wise or slice-wise training are often employed to fit a 3D dataset within these parameter-heavy models under limited computer memory, sacrificing fi-ne-scale geometric information from input images and potentially affecting clinical diagnoses [1-3]. Since the MRI dataset has different spatial resolutions in the third dimension, 3D-MRI images are flattened into 2D slices.”
Pages 2& 3
|
|
6 |
It’s unclear to me but I think it’s important to clarify, at conceptual level, what you’re relying on to decide if a pixel belongs to a tumor or not. Pixel values, or pixel value deviation from the norm, pixel grouping according to some criteria, or something else? Lines 403-406 appear to suggest that pixel value is the key. If so, do you really need to use all those sophisticated techniques to identify the tumor pixels, particularly if you’re only. |
OK, in general level set algorithm follows region growing (also called region merging) that is a technique for extracting a connected region of the image which consists of groups of pixels/voxels with similar intensities [64]. In its simplest form, region growing starts with a seed point (pixel/voxel) that belongs to the object of interest. The seed point can be manually selected by an operator or automatically initialized with a seed finding algorithm. Then, region growing examines all neighboring pixels/voxels and if their intensities are similar enough (satisfying a predefined uniformity or homogeneity criterion), they are added to the growing region. This procedure is repeated until no more pixels/voxels can be added to the region.
An important factor in reducing segmentation error and the number of required iterations when using the level set technique is the choice of the initial contour points, both of which are important when dealing with the wide range of sizes, shapes, and structures that brain tumors may take. To define the velocity function, conventional methods simply use the image gradient, edge strength, and region intensity. This article suggests a clustering method influenced by the Quantum Inspired Dragonfly Algorithm (QDA), a metaheuristic optimizer inspired by the swarming behaviors of dragonflies, to accurately extract initial contour points. An initial contour for the MRI series will be derived from these extracted edges. The final step is to use a level-set segmentation technique to isolate the tumor area across all volume segments.
In the revised version, subsection 1.2. "Research Contribution and Novelty," was modified to illustrate the main contribution of the suggested model and clarify its need to deal with the difficulties faced by state-of-the-art region growing-based approaches.
“In this work, a modified level-set segmentation method based on a quantum-inspired optimization technique is suggested to detect tumors in 3D-MRI scans of the brain. The initial position is very important in managing topological variations in contours due to the tumor's size, shape, and structural variability. To get the precise initial contour points for the level-set segmentation method, the proposed approach employs an enhanced clustering strategy that integrates k-means and the quantum version of the dragonfly algorithm (QDA), in which the k-means are employed to determine the initial position of the QDA’s population centroid, rather than using a random initial position. The preprocessing stage involves removing the brain from the cranium, and the two-step QDA is used to extract the tumor edges. As an initial con-tour for the MRI scan, these edges will be employed. Using a level-set segmentation technique, the tumor location will then be detected from all volume slices. In this study, for optimization in the selection of the initial contour points, dragonfly behaviors are quantimized using quantum computing to improve the DA's search efficiency. To go from a local optimal solution to a better solution (escape from the local solution) and hopefully to the global optimum, each dragonfly may employ a quantum rotation gate to overcome the inertia weight during the search operations and to conquer the searching quality of Levy flight” |
|
6 |
Figure 8: not my field so I don’t really know. But I’m just wondering, the original images would undergo a lot of changes to make them segmentation friendly. Can one still trust results from the somewhat “doctored” images? |
Yes, the original images would undergo a lot of changes to make them segmentation-friendly, and the preprocessing stage is necessary to achieve high-accuracy results.
Still, anyone can trust the results by identifying the image's source and analyzing it for doctoring with advanced tools.
In the literature, there are many algorithms that can easily differentiate between benign (e.g., noise removal) and malicious modifications (“doctored” images). |
|
7 |
Figure 6: wouldn’t Gaussian filtering change/hide details of features of interest? |
No, because the kernel is not hard towards drastic color change (edges) due to the fact that the pixels towards the center of the kernel have more weightage towards the final value than the periphery. The role of sigma in the Gaussian filter is to control the variation around its mean value. So as the Sigma becomes larger the more variance allowed around mean and as the Sigma becomes smaller the less variance allowed around mean.
In the revised version, the Figure 6 caption was modified to illustrate this fact.
“Figure 6: (a) Synthetic MR brain image, axial section, maximum intensity noise (5%); (b) image filtered with fixed Gaussian window size; (c) image filtered with decreasing window size at the same number of iterations. A Gaussian Filter is a low-pass filter used for reducing noise (high-frequency components). The kernel is not hard on drastic color changes (edges) due to the pixels towards the center of the kernel having more weightage towards the final value than the periphery”.
|
|
8 |
In Section 5, memory usage was indicated. Be nice to also let the reader to have a feel about the runtime (seconds, mins, hours or days)? |
In the revised version, a new set of experiments was conducted to calculate the running time of the suggested model to segment one 3D MRI volume. A new sentence was added to clarify the runtime needed to segment one 3D MRI volume.
“The execution speed was quite good, with an average of less than 7 seconds per volume to test, i.e., 45 ms per image.”
“Herein, each 3D volume includes 155 2D slices/images of brain MRIs collected at various locations across the brain. Every slice is 240×240 pixels in size in NIfTI format and is made up of single-channel grayscale pixels. With NIfTI files, images and other data are stored in a 3D format. It's specifically designed this way to overcome the spa-tial orientation challenges of other medical image file formats”.
Page 26 |
|
9 |
In conclusions, after all the troubles, improvement as stated was 2.5%. - Comment on significance of this figure should help the reader to better appreciate the effort and importance. - - Also state the baseline, i.e., 2.5% on top of what value, 96.2%? Are we chasing a diminishing return here? The method hasn’t been tested for multiple tumor lumps per slice. This is a serious limitation, isn’t it? Because a single cancerous mass may show up as separate lumps in a slice. Also what does it mean in practice? Someone human needs to make a judgment that there is only one tumor to look for before they feed the data to your model to quantify it? |
In the revised version, more sentences were added to comment on the significance of the suggested model compared to state-of-the-art brain segmentation approaches.
“One potential reason for this result is that a two-step QAD identifies a more accurate initial contour than the comparison approaches do, thereby increasing the segmentation accuracy. Information gathered from k-means is used to support QDA's search engine. By using a quantum-inspired computing paradigm, the suggested model is able to stabilize the exploitation/exploration trade-off and make up for any weaknesses of the conventional DA-based clustering approach, such as delayed convergence or being stuck in a local optimum. Furthermore, the suggested model has a robust local search capacity by adopting a mutation procedure to enhance the swarm's mutating and realizing its variety. Figure 18 shows a sample of segmentation results”
“Even if the results are more or less in agreement with these approaches. Overfitting is the primary issue with CNN, and the need for a large training set makes it computationally costly”.
“The convergence rate in QDA is parameter-free. The suggested model has a robust local search capacity by adopting a mutation procedure to enhance the swarm's mutating and realizing its variety”
“It can be deduced that the QDA method significantly decreases the required number of level set iterations. The contour was calculated very closely to the tumor region using the best-configured two step QDA parameters”.
In the revised version, a new set of experiments was conducted to validate the efficiency of the proposed model when dealing with other types of tumor types with multiple tumor lumps per slice.
Figure 20 – Page 27 Table 6 -Page 26
|
Round 2
Reviewer 1 Report
The authors have revised the manuscript significantly.
I just curious on the introduction:
According to the American National Brain Tumor Society, there were around 700,000 people living with brain tumors in the United States in 2017 [1].
Why the data is only for USA? Maybe can add other statistics too.
The rest is good.
Author Response
To The Editor-in-chief and the Associate Editor,
Bioengineering.
|
Title: A New Medical Analytical Framework for Automated Detection of MRI Brain Tumor using Evolutionary Quantum Inspired Level Set Technique
Dear Sir,
Thanks for giving us the opportunity to revise the paper for the possible publication in your interesting journal. We also thank the reviewers for their constructive comments. We have significantly revised the manuscripts based the comments received and the details are summarized below.
Reply to the comments:
In the revised version, what was modified and added was written in red
Reviewer 1
|
S. No |
Comments |
Changes made in the revised manuscript based on the comments |
|
1 |
The authors have revised the manuscript significantly. |
Thank you for the positive feedback. |
|
2 |
I just curious on the introduction:
According to the American National Brain Tumor Society, there were around 700,000 people living with brain tumors in the United States in 2017 [1].
Why the data is only for USA? Maybe can add other statistics too. |
Ok, in the revised version, more recent statistics were added, according to the World Health Organization website.
“Brain tumors account for 85% to 90% of all primary central nervous system (CNS) tumors. Worldwide, an estimated 308,102 people were diagnosed with a primary brain or spinal cord tumor in 2020. In 2023, an estimated 24,810 adults (14,280 men and 10,530 women) in the United States will be diagnosed with primary cancerous tumors of the brain and spinal cord [1].”
|